# Synergetic regulation of SEI mechanics and crystallographic orientation for stable lithium metal pouch cells

Yanhua Zhang[1], Rui Qiao[1], Qiaona Nie[1], Peiyu Zhao[1], Yong Li[2], Yunfei Hong[1], Shengjie Chen[1], Chao Li[1,3], Baoyu Sun[1], Hao Fan[1], Junkai Deng [1], Jingying Xie[2], Feng Liu[1] & Jiangxuan Song [1] ✉

The advancement of Li-metal batteries is significantly impeded by the presence of unstable solid electrolyte interphase and Li dendrites upon cycling. Herein, we present an innovative approach to address these issues through the synergetic regulation of solid electrolyte interphase mechanics and Li crystallography using yttrium fluoride/polymethyl methacrylate composite layer. Specifically, we demonstrate the in-situ generation of Y-doped lithium metal through the reaction of composite layer with Li metal, which reduces the surface energy of the (200) plane, and tunes the preferential crystallographic orientation to (200) plane from conventional (110) plane during Li plating. These changes effectively passivate Li metal, thereby significantly reducing undesired side reactions between Li and electrolytes by 4 times. Meanwhile, the composite layer with suitable modulus (~1.02 GPa) can enhance mechanical stability and maintain structural stability of SEI. Consequently, a 4.2 Ah pouch cell with high energy density of 468 Wh kg$^{-1}$ and remarkable capacity stability of 0.08% decay/cycle is demonstrated under harsh condition, such as high-areal-capacity cathode (6 mAh cm$^{-2}$), lean electrolyte (1.98 g Ah$^{-1}$), and high current density (3 mA cm$^{-2}$). Our findings highlight the potential of reactive composite layer as a promising strategy for the development of stable Li-metal batteries.

Newly emerging applications, such as electric vehicles and large-scale smart grids, are in dire need of high-specific-energy and long-cycling rechargeable batteries[1,2]. Lithium metal, with high theoretical specific capacity (3860 mAh g$^{-1}$) and low electrode potential (−3.04 V, vs. standard hydrogen electrode), is recognized as an ideal anode material[3,4]. To achieve high-energy-density lithium metal batteries (LMBs) beyond 400 Wh kg$^{-1}$, up to 500 Wh kg$^{-1}$, harsh conditions including high cathode capacity (>4 mAh cm$^{-2}$), low negative/positive ratio (N/P ratio) <2, and restricted electrolyte amount/capacity ratio (E/C ratio) <3 g Ah$^{-1}$ are necessary[5]. Unfortunately, highly reactive Li

metal easily reacts with liquid electrolytes and forms an unstable solid electrolyte interphase (SEI). The unstable SEI leads to uneven Li plating and stripping, which results in low Coulombic efficiency (CE) and rapid capacity fading[6,7]. These problems severely impede the achievement of energy target especially under stringent conditions.

Therefore, constructing a stable SEI on Li metal anodes is necessary for practical LMBs. Effective strategies to enhance the SEI stability of Li metal anodes focus on liquid electrolyte design[8–12] and artificial protective layers[13,14]. Recently, Li et al. reported that SEI layer controls the Li$^+$ transport and ultimately affects the deposition morphology under

[1]State Key Laboratory for Mechanical Behavior of Materials, Shaanxi International Research Center for Soft Matter, Xi'an Jiaotong University, Xi'an 710049, China. [2]State Key Laboratory of Space Power-Sources Technology, Shanghai Institute of Space Power-Sources, Shanghai 200000, China. [3]Instrumental Analysis Center, Xi'an Jiaotong Univerusity, Xi'an 710049, China. ✉e-mail: songjx@xjtu.edu.cn

normal current density ($-1$ mA cm$^{-2}$)[15]. These results demonstrate that a stable SEI plays a vital role to regulate Li$^+$ transport and deposition morphology. Although the considerable success achieved in stabilizing SEI to a certain extent at low areal capacities (<2 mAh cm$^{-2}$), deposition morphologies under high areal capacity (≥4 mAh cm$^{-2}$) are filaments, nanorods, columns or chunks[15], and tend to pack up into loose and porous structure during the growth process, which not only leads to severe side reactions between Li metal and electrolytes due to the large specific surface area, but also results in huge volume expansion during repeated cycles[16,17]. The main reason for this phenomenon is that the reactivity of Li metal with electrolytes is not effectively suppressed, that is the preferred orientation of Li plating is not fundamentally regulated, which is tightly related to the deposition morphology[18]. Thus, forming a dense deposition morphology under high areal capacity (≥4 mAh cm$^{-2}$) is much essential and still highly challenging, which is beneficial for realizing the goal of high-energy-density LMBs.

Undoubtedly, the composition of SEI also plays an extremely important role on the deposited morphology[16,19,20]. The SEI formed in the electrolyte is inhomogeneous due to its complex and uncontrollable composition, which leads to uneven Li$^+$ distribution and migration, thereby causing non-uniform and porous deposition morphology, especially under high areal capacity[21]. Modified SEI is an essential way to stabilize Li metal anode. To date, several functional artificial SEI layers have been studied, involving inorganic SEI layer, organic SEI layer, organic-inorganic hybrid SEI layer[21]. It is well known that LiF with high interfacial energy, high chemical stability, and low Li$^+$ diffusion barrier is generally regarded as an ideal SEI component to regulate Li deposition[22]. Therefore, constructing a LiF-rich SEI on the Li metal is a common strategy, and can promote uniform deposition[23–26]. However, the LiF-rich SEI is brittle, and has poor mechanical stability, which facilitating its rupture and causing huge volume fluctuation during repeated cycles. Organic components in SEI are generally flexible and viscoelastic, which can withstand volumetric deformation. Whereas, most of the current polymers hardly inhibit dendrite growth. Therefore, the organic-inorganic hybrid SEI is crucial to enable high-performance LMBs[16,19]. Although promising process, this structure of SEI still fails to induce dense deposition of Li on current collector, especially at high areal capacity.

Based on the above discussions, a SEI suitable for high-energy-density LMBs should simultaneously satisfy two critical conditions: (1) dense deposition in the bottom layer to relieve the reactivity between Li metal and electrolytes; (2) mechanical stability in the top layer to accommodate volume expansion. Although the currently reported SEI could improve cycling stability to a certain extent at low areal capacities (<2 mAh cm$^{-2}$), achieving both requirements of dense deposition and mechanical stability simultaneously at high areal capacities (≥4 mAh cm$^{-2}$) remains a significant challenge for the future goal of higher-energy-density and long-cycle-life LMBs.

Herein, we report a synergetic regulation strategy of SEI mechanics and crystallographic orientation of Li via a YF$_3$/polymethyl methacrylate composite (YP) layer (Fig. 1). Among them, a Y-doped Li metal can be generated by the reaction of YP with Li metal, and Y-doped (200) plane has the lowest surface energy, which enables the preferred oriented growth of Li metal along the (200) crystal plane rather than conventional (110) plane and relieves the reactivity between Li metal and electrolytes during cycling, ultimately accomplishing dense and smooth deposition even at high-areal-capacity of 10 mAh cm$^{-2}$. Moreover, the formed LiF-rich SEI leads to rapid charge transfer kinetics. Meanwhile, YP layer with a suitable modulus of $-1.02$ GPa serves as a protective sheath to enhance mechanical stability and uniform stress distribution of SEI, thus preventing its structure from being destroyed upon cycling. More attractively, a 4.2 Ah Li metal pouch cell, under a low N/P ratio of 1.67, lean electrolyte of 1.98 Ah g$^{-1}$, and high current density of 3 mA cm$^{-2}$, reaches a high energy density of 468 Wh kg$^{-1}$ and undergoes 150 cycles.

## Results

### In situ construction and characteristics

Achieving dense deposition at the bottom and mechanical stability at the top are critical for high-areal-capacity Li metal anodes to meet high-energy-density LMBs. At this point, we aimed to construct a composite protective layer on Li metal anodes that can relieve the reactivity between Li metal and electrolytes in the bottom layer and enhance mechanical stability of SEI in the top layer upon cycling. Therefore, regulating Li metal's preferential growth orientation and building a stable SEI on Li metal anode are necessary for realizing high-energy-density LMBs. The preferred crystallographic orientation depends on the surface energy of the crystal[27,28]. Doping with rare-earth element is considered as an efficient strategy to change the surface energy and consequently regulate the preferred crystallographic orientation[29]. The in-situ reaction between YF$_3$ and Li metal results in the generation of Y-doped Li metal and LiF. Y doping into Li metal could tune and reduce the surface energy of Li metal, and thus regulate the preferred orientations, ultimately reducing undesired side reactions between Li metal and electrolytes and inhibiting Li dendrite growth. Meanwhile, the formed LiF-rich SEI could accelerate charge transfer kinetics. Furthermore, polymethyl methacrylate (PMMA), with excellent polymeric segmental motion, high ionic conductivity, and elastic modulus[30], can enhance mechanical stability and uniform stress distribution of SEI, thus preventing its structure from being destroyed upon cycling and contributing to the formation of a stable SEI.

Based on the aforementioned hypothesis, we propose a synergetic regulation strategy of SEI mechanics and crystallographic orientation of Li through a YF$_3$/PMMA composite layer (denoted as YP). As observed, the thickness of the YP layer is $-2.42$ μm (Supplementary Fig. 1). The uniform distribution of C, O, Y, and F elements further proves that YP are evenly covered on the Cu foil (Supplementary Fig. 2). The C=O, C-O, and -CH$_2$ stretching bands at $-1736$, 1130 and 1074 cm$^{-1}$ in Fourier transform infrared (FTIR) verify the existence of PMMA (Supplementary Fig. 3). Moreover, 2D Raman depth-profiling mapping, tested at 2951 cm$^{-1}$ referring to C-H stretching in PMMA, indicates that the PMMA mainly focused on the surface of deposited Li metal (Supplementary Fig. 4).

In addition, X-ray photoelectron spectroscopy (XPS) was further carried out for YP-Cu electrode. Before cycling, the binding energy at 284.6, 285.9, and 288.8 eV refer to the C-C, C-O, and C=O bonds of PMMA (Supplementary Fig. 5a), respectively. The binding energy at $-159.8$ eV in the Y $3d_{5/2}$ spectra is assigned to YF$_3$ (Supplementary Fig. 5b), signifying the successful coating of YP on the Cu foil. After Li deposition, the binding energy at $-157.8$ eV in the Y $3d_{5/2}$ spectra (Supplementary Fig. 5e) and 55.8 eV in the Li $1s$ spectra (Supplementary Fig. 5f) correspond to Y and LiF[14], indicating the reduction of YF$_3$ to form Y and LiF during Li plating. Moreover, Cryo-transmission electron microscopy (Cryo-TEM) was also employed to further demonstrate the Y doping into the Li metal. HRTEM images reveal a lattice space of 0.24 nm corresponding to the (110) plane of Li crystal (Supplementary Fig. 6a, b), as verified by the corresponding local Fourier transform images (inset of Supplementary Fig. 6c). Energy-dispersive X-ray spectroscopy (EDS) elemental mapping results (Supplementary Fig. 6d) indicate the existence of Y element in the lithium metal, providing additional confirmation of Y doping into the Li metal. Therefore, the combined results from Cryo-TEM and XPS suggest that Y has been successfully doped into Li metal.

### Analysis of the preferred orientation growth and interfacial stability

In situ X-ray diffraction (XRD) was employed to dynamically monitor the crystal plane evolution with and without YP layer after the first Li plating (Fig. 2a, b and Supplementary Figs. 7, 8). Typical discharge curves of the bare Li and YP-Li in the first cycle are shown in Supplementary Fig. 7a, c. Li metal is a body-centered cubic structure, and the

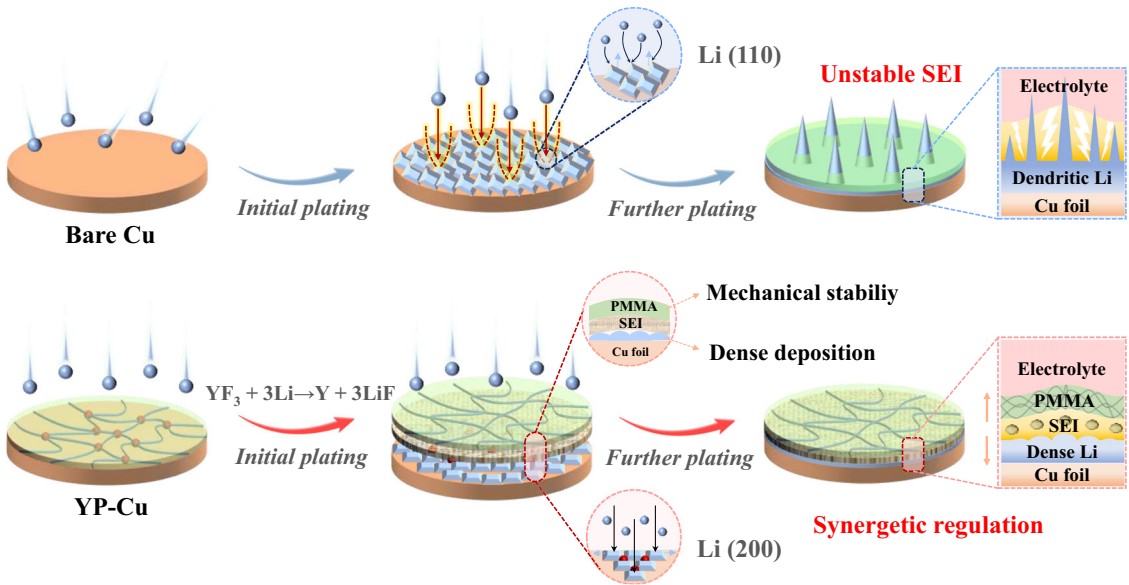

**Fig. 1 | Schematic of lithium plating behavior on Cu foil with and without YP layer.**

(110), (200), and (211) crystal planes are the dominant crystallographic features in the XRD pattern[31]. Owing to the most densely packed and the lowest-energy surface of the (110) plane, Li metal typically grows along the (110) plane during plating process and the (110) crystal plane is regarded as the preferential growth crystal plane, resulting in uncontrollable dendrite growth and unstable interface[32,33]. As expected, bare Li exhibits a dominant XRD peak belonging to the (110) crystal plane at the early stage of Li plating (Supplementary Fig. 7b, e). Then, the peak of (110) plane tend to become intensified upon prolonging Li plating time (Fig. 2a). This indicates that (110) plane is the main exposed crystal plane. In comparison, for the YP-Li electrode, the peak intensity corresponding to the (200) plane increases significantly during Li deposition (Fig. 2b and Supplementary Fig. 7d, f), suggesting that (200) plane is the preferred orientation[34]. Thus, in situ XRD can effectively detect a shift of the preferred orientation from the (110) plane to (200) plane when Y doped into Li metal during first Li plating. Synchronously, ex situ XRD was also scrutinized to track the crystal plane evolution after different cycles, as shown in Supplementary Fig. 9. Bare Li shows a dominant XRD peak belonging to the (110) crystal plane after different cycles (Supplementary Fig. 9a). While the (200) plane becomes dominant for the YP-Li electrodes, which indicates a change from (110) to (200) in the preferred orientation upon cycling (Supplementary Fig. 9b). To further exclude the effect of PMMA on the preferred orientation, the XRD of the PMMA on the Li surface after different cycles was also examined and the transformation in preferred orientation was not detected (Supplementary Fig. 10), implying that the preferred orientation stems from Y doping instead of PMMA.

Density functional theory (DFT) calculation was then conducted by calculating the surface energy for three crystal planes to further gain insight into the underlying mechanism of the preferred orientation[35]. The structure, atomic position and lattice constant of Li slabs for (200), (110), and (211) plane with and without YP layer are displayed in Fig. 2c when Y doped to the surface layer of Li metal. The calculated results of surface energies are summarized in Fig. 2d and Supplementary Table 1. Obviously, the relationship among these planes for bare Li is $\bar{E}_{S-Li(110)} < \bar{E}_{S-Li(211)} < \bar{E}_{S-Li(200)}$, indicating that (110) plane is the most stable surface than others. indicating that (110) plane is thermodynamically favorable for nucleation and growth of Li metal. Surprisingly, the Y doping can tune the energy of these planes. Apparently, the energy relationship is changed to $\bar{E}_{S-Li_{95}Y(200)} <$

$\bar{E}_{S-Li_{95}Y(110)} < \bar{E}_{S-Li_{95}Y(211)}$, meaning that (200) plane has the lowest surface energy, even lower than any low-surface-energy crystalline planes of the Li metal. This result suggests that Y doping intrinsically reduces the surface energy, enabling the (200) plane becomes the preferred orientation[36]. Moreover, a similar conclusion is also obtained for the surface energy when Y doped on the second layer of Li metal (Supplementary Fig. 11).

Owing to the exposure of the lowest-surface-energy crystal plane to the electrolyte, the reactivity of lithium metal towards the electrolyte is suppressed[37], which was also demonstrated by the potentiodynamic polarization experiments after different cycles. Supplementary Figs. 12, 13 and Fig. 2e show that YP-Li demonstrates a low and stable corrosion current density (~0.30 mA cm$^{-2}$) during whole cycling process (Supplementary Fig. 12), which suggests that YP-Li exhibits a low extent of side reactions during cycling. While the corrosion current density of bare Li experiences a substantial increase (from 0.62 to 1.42 mA cm$^{-2}$), indicating that YP can relieve the reactivity between Li metal and electrolytes during cycling. To further investigate whether the reduced side reactions is attributed to the Y doping or PMMA, we also designed a control sample with only PMMA on the Li surface, where the corrosion current density of PMMA-Li is 0.55, 0.64, and 0.89 mA cm$^{-2}$ after 1, 20, and 50 cycles, respectively (Supplementary Fig. 13). These values are lower than that of bare Li but higher than YP-Li (Fig. 2e), which is attributed to the protection of PMMA. This protection contributes to shielding the lithium metal from undesirable side reactions with electrolytes[30]. Therefore, it can be concluded that the synergistic effect of Y-doped Li metal and PMMA protective layer reduces the reactivity of Li metal with electrolytes, thus enabling a stable lithium-electrolyte interface.

## Structure and components characterization of SEI

The stability of high-areal-capacity Li metal is critically associated with a stable SEI. The LiF-rich SEI exhibits tremendous potential in prolonging

the cycling lifespan of the Li metal anode by suppressing continuous side reactions and enabling planar Li deposition rather than Li dendrite growth[23,38]. More importantly, the advantage of YP layer in our study is the spontaneous generation of LiF by in-situ chemical/ electrochemical reduction of YF$_3$ with deposited Li metal. Here, the nanostructure and component of SEI were also investigated in virtue of Cryo-TEM characterization. As observed, the SEI layers formed with

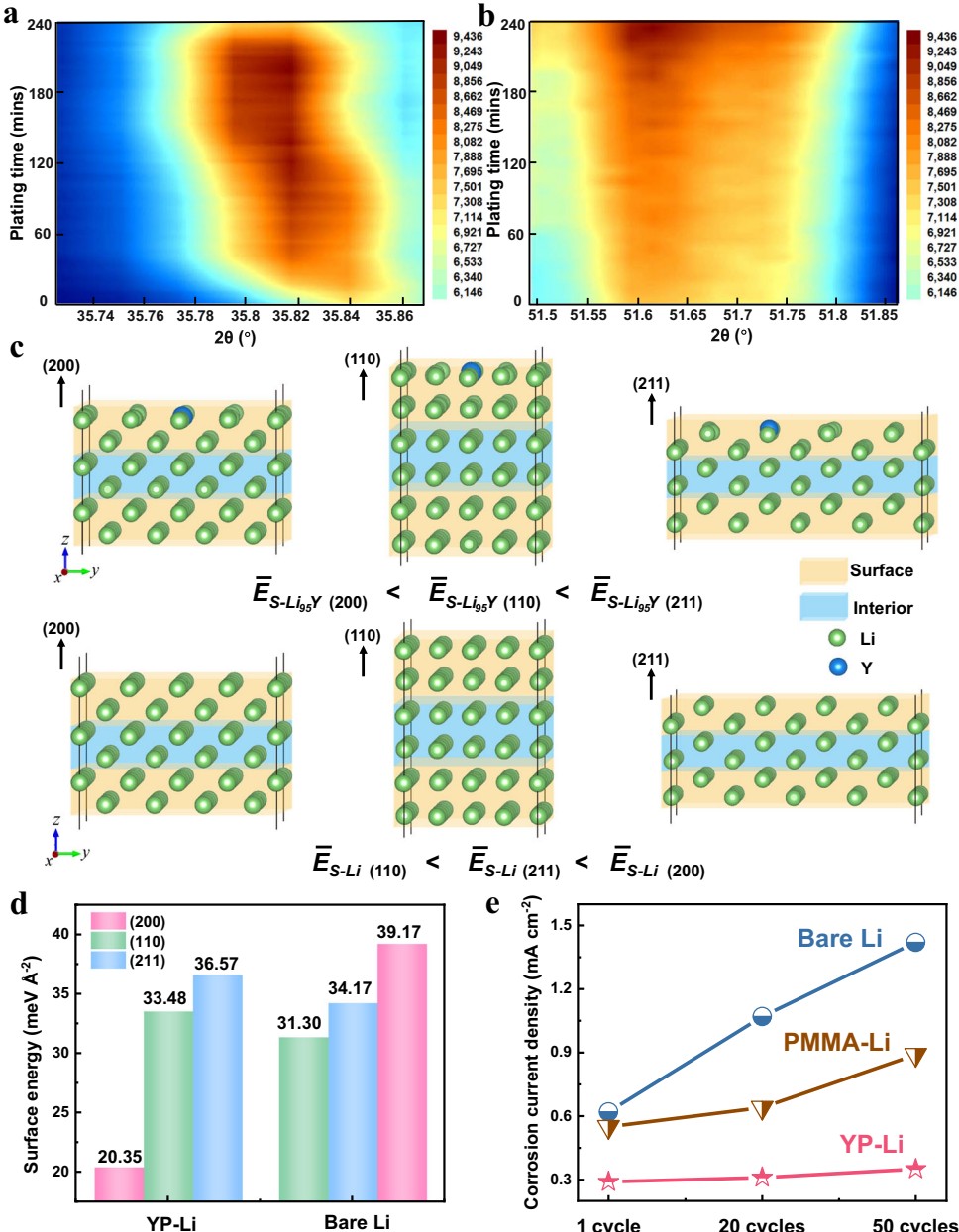

**Fig. 2 | Characteristics and mechanism of the preferred orientation growth.** In situ XRD contour map during Li deposition for **a** the (110) crystal plane of the bare Li, and **b** the (200) crystal plane of the YP-Li. **c** The structure, atomic position, and lattice constant of Li slabs with and without YP layer in the fully relaxed configuration at (200), (110), and (211) crystal planes. **d** Surface energies with and without YP layer for (200), (110), and (211) planes. **e** The corrosion current density of YP-Li, PMMA-Li, bare Li electrodes after 1, 20, and 50 cycles.

and without YP are completely different. The formed SEI induced by YP exhibits a mosaic structure that consists of an amorphous polymer matrix and embedded LiF-enriched nanocrystals (Fig. 3a). The corresponding local fast Fourier transform images clearly recognize the lattice spacings of the crystalline components in the six regions, where the lattice spacings of $Li_2O$ (311) plane, LiF (200) plane, and LiF (111) plane are 0.14 nm, 0.20 nm, and 0.23 nm, respectively (Supplementary Fig. 14)[39]. Notably, YP-Cu exhibits smooth lithium surface with a thin SEI layer of ~20 nm, which is beneficial for enabling uniform and dendrite-free Li deposition morphology even at high areal capacity. In contrast, bare Cu shows amorphous SEI morphology, where dendrite morphology can be clearly detected (Supplementary Fig. 15). The more detailed composition and structure of SEI were further performed by Time-of-Flight Secondary Ion Mass Spectrometer (ToF-SIMS) based on

the depth profiles and 3D views of several fragments (Fig. 3b and Supplementary Fig. 16). Here, $LiF_2^-$, $CH_3O^-$, and $C_2H_2O^-$ are characteristic ionic fragments of LiF and organic components, respectively. The result of ToF-SIMS is consistent with that of Cryo-TEM. Specifically, LiF is abundant in the YP derived SEI, but bare Cu contains more organic components than YP derived SEI, implying that YP can induce the generation of more LiF component, which greatly facilitates uniform lithium deposition and prevents parasitic reactions.

XPS measurements were further applied to verify the stability of SEI under a high areal capacity of 4 mAh cm⁻² at 1 mA cm⁻² (Supplementary Fig. 17-18). In the C 1 *s* spectra, we can detect that C-C, C-O and $Li-CO_2$ peaks are on the surface of YP-Cu with negligible changes after both 1 and 20 cycles. However, there appears a new peak at ~289.6 and 290.9 eV for bare Cu after 20 cycles, indexed to $COO^-$ and $CO_3^{2-}$,

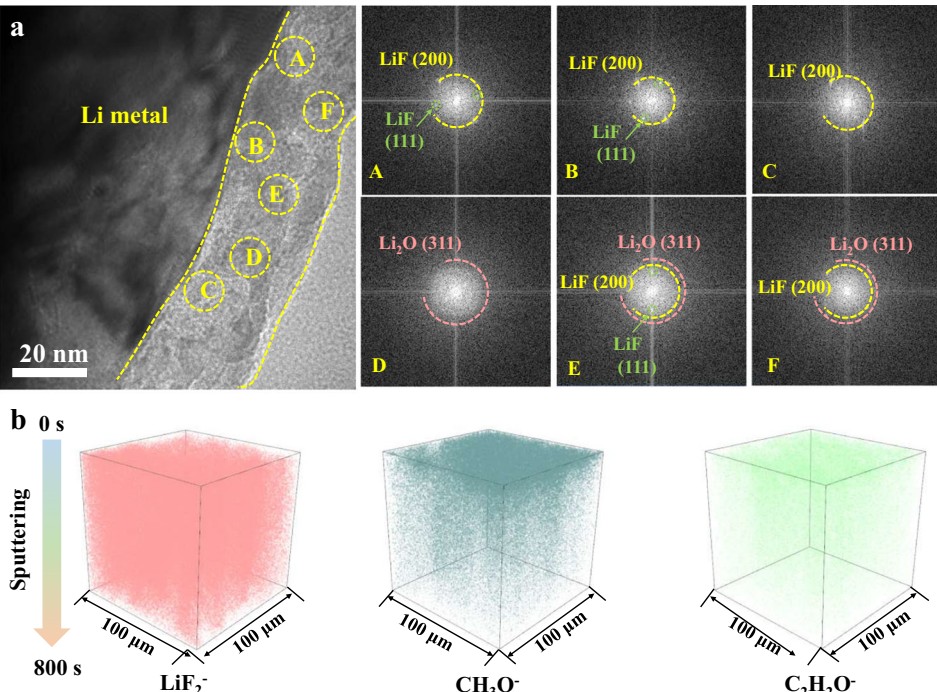

**Fig. 3 | Structure and components characterization of SEI. a** Cryo-TEM images of the YP induced SEI layer, and the corresponding local fast Fourier transform images. **b** The 3D views of $LiF_2^-$ (for LiF), $CH_3O^-$, and $C_2H_2O^-$ (for organic components) in the ToF-SIMS sputtered volumes of the YP-Cu induced SEI.

indicating the electrolyte depletion during cycling due to the destroy of SEI layers[40]. In the Li 1$s$ spectra, the intensity of LiF (at ~55.9 eV) in YP-Cu is higher than that of bare Cu, indicating more LiF component. In addition, the Li-O bond ascribed from the strong interactions between the carbonyl of PMMA and Li$^+$ is also detected, which could regulate uniform Li deposition[30]. Even after 20 cycles, the intensities of LiF, Li-CO$_2$, and Li-O groups are almost the same with after 1 cycle in the YP-Cu electrode, while the intensity of LiF obviously decreases after 20 cycles, suggesting the unstable SEI for bare Cu.

## Analysis of mechanical stability of SEI

Furthermore, YP-Cu plays a vital role in ensuring the mechanical stability of SEI. As illustrated in atomic force microscope (AFM), YP has a mean Young's modulus of 0.83 GPa (Supplementary Fig. 19), which can adapt to volume expansion upon cycling. The modulus was further measured by AFM to quantify the mechanical stability of interface for YP-Cu, as shown in Fig. 4a. The YP layer exhibits ~0.94 GPa after 1 cycle and remain almost constant even after 20 cycles, meaning excellent tolerance to the volumetric deformation and enhanced mechanical stability upon cycling. In addition, the mean modulus of SEI is ~4.06 GPa after 1 cycle and retains ~4.32 GPa even after 20 cycles, indicating the dendrite growth is suppressed[41]. To gain insight into the underlying mechanism of YP-Cu during cycling, the finite element method (FEM) simulations were conducted to investigate the stress distribution. At the early stage (0 s), both of them show negligible and small stress value (Supplementary Fig. 20). As the cycling times up to 3000 s, the phenomenon of inhomogeneous stress starts to appear for bare Cu. And the stress nonuniformity tends to be more pronounced at the top of SEI when cycling to 10000 s (Fig. 4c). That is because that the uneven Li$^+$ deposition and huge volume expansion lead to the formation of the stress field, and the release of stress causes the crack of SEI around the large stress region. Then, lithium metal grows from the crack, aggravating the dendrites growth. In stark contrast, the stress of YP-Cu displays uniform and small level, generating a more homogeneous and mechanically stable SEI (Fig. 4b). The above results further demonstrate that YP-Cu can induce dense lithium deposition

and alleviate volume expansion, ultimately enabling a mechanically stable SEI.

Besides, the interfacial properties and kinetics behavior can also affect the Li deposition behavior. The activation energy (E$_a$) for Li ion diffusion across SEI layer was further proved via fitting the temperature-dependent electrochemical impedance spectroscopy (EIS) profiles with Arrhenius equation (Supplementary Fig. 21). The E$_a$ for YP-Cu and bare Cu are calculated to be 2.591 and 5.338 kJ mol$^{-1}$, respectively. Such evidently decreased E$_a$ demonstrates a smaller barrier and rapid Li-ion diffusion through the interface for YP-Cu. To further verify the significant role of YP in stabilizing the SEI and improving the kinetics upon cycling, EIS measurements were conducted after different cycles with high areal capacity of 4 mAh cm$^{-2}$ at 1 mA cm$^{-2}$. As plotted in Supplementary Fig. 22, YP-Cu displays stable and small charge transfer resistance of ~16 Ω even after 50 cycles, while a dramatic resistance swing is found in bare Cu cells (from 18 to 42 Ω). These phenomena support the advantage of the YP-Cu electrode in stabilizing the lithium/electrolyte interface and enhancing Li plating/stripping kinetics, which contributes to uniform, smooth and dense lithium deposition morphologies.

## Deposited morphologies and cycling performances of the asymmetric Li||Cu cells at different areal capacities

The evolution of the Li deposition behavior at different areal capacity with a fixed current density of 1 mA cm$^{-2}$ after the first cycle was then investigated to further highlight the superiority of YP layer (Fig. 5). At the early stage (1 h, 1 mAh cm$^{-2}$), the deposited Li shows a sheet-like structure (Fig. 5a), and gradually grows along this sheet-like structure and continuously fills the voids as the deposition capacity increases to 4 mAh cm$^{-2}$ (Fig. 5b). When further increasing to 10 mAh cm$^{-2}$, the plating Li presents a dense and flat morphology (Fig. 5c). In sharp contrast, Li deposition on the bare Cu starts wirh a mossy dendritic structure, and the lithium dendrites gradually become longer and coarser as the increase of deposition capacity from 1 to 10 mAh cm$^{-2}$, resulting in porous and loose deposition morphologies (Fig. 5g–i). At the Li stripping state, no obvious Li residues are observed on the

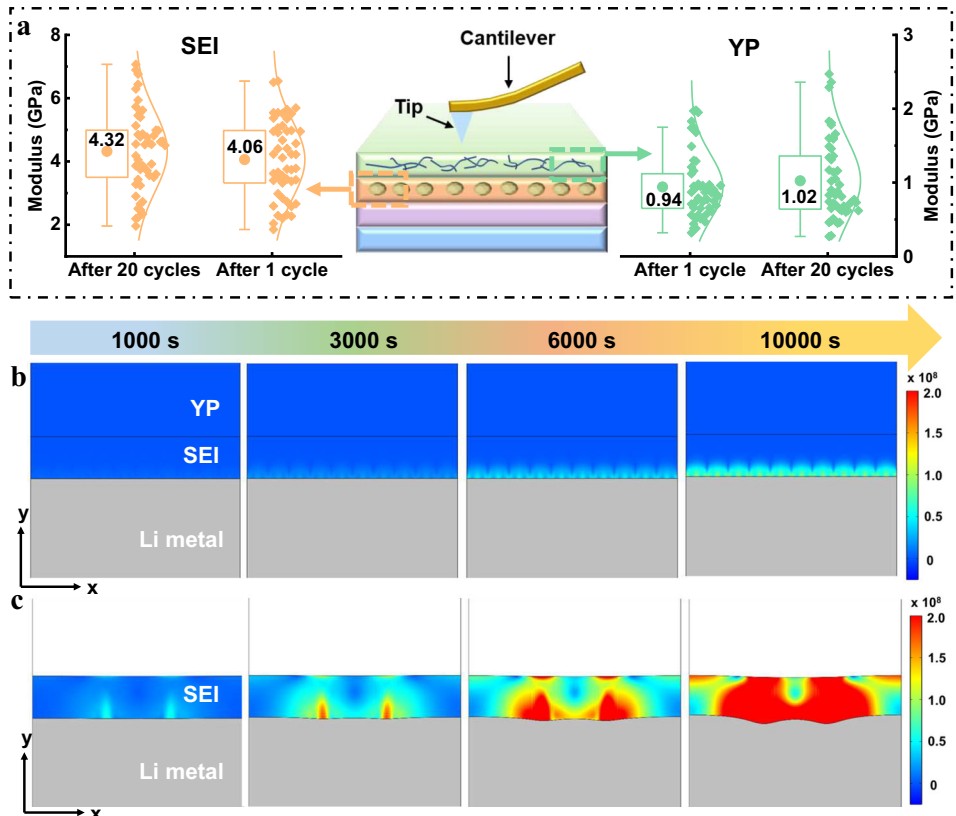

**Fig. 4 | Analysis of the mechanical stability. a** The distribution of DMT modulus for the SEI and YP layer of YP-Cu electrode after 1 cycle and 20 cycles. Stress distribution of SEI during cycling for **b** YP-Cu and **c** bare Cu electrodes.

surface of YP-Cu electrodes. However, a large amount of Li residues appears on bare Cu electrodes (Supplementary Figs. 23–25), indicating that the lithium metal anode could not undergoes reversible plating/ stripping without YP. With the increase of cycling life, the deposited Li on the YP-Cu electrode still maintains an even and intact dendrite-free morphology after 20 (~160 h) and 40 cycles (~320 h) at 4 mAh cm$^{-2}$. On the contrary, bare Cu appears severe cracks (Supplementary Fig. 26), which seriously affects lithium ion and electron transport. Even after Li stripping for 20 and 40 cycles, YP-Cu also shows a smooth surface in comparison with the bare Cu electrode (Supplementary Figs. 27 and 28), further implying the vital role of the YP layer in stabilizing SEI during repeated Li plating/stripping process.

Meanwhile, the thickness of the deposited Li was also measured to quantitatively evaluate the volumetric stability during cycling. As shown in Fig. 5d–f, for the YP-Cu electrode, the deposited Li shows thickness of ~6.34, 20.96, and 50.60 μm at deposited capacity of 1, 4, and 10 mAh cm$^{-2}$, respectively. These thicknesses are close to the theoretical value of 5, 20, and 50 μm for 1, 4, and 10 mAh cm$^{-2}$[42]. In comparison, for the bare Cu electrode, the Li deposition thickness is 14.35, 32.34, and 97.13 μm at deposited capacity of 1, 4, and 10 mAh cm$^{-2}$, respectively (Fig. 5j–l). With the extension of the cycling, the morphology of Li plated on the YP-Cu electrode is uniform and dense, showing the thickness of 22.76 and 24.62 μm after 20 and 40 cycles. Whereas, the thickness of Li plating increases dramatically to 49.71 and 79.39 μm on the bare Cu (Supplementary Fig. 29), which is two times thicker than that of the YP-Cu electrodes. The sharp contrast verifies that the YP layer can efficiently induce dense deposition and alleviate volumetric changes upon cycling.

The stability of Li plating/stripping was further explored for assembling asymmetrical Li||Cu cells with different areal capacities at a fixed current density of 1 mA cm$^{-2}$. Under the plating capacity of 2 mAh cm$^{-2}$, YP-Cu displays a more stable cycling performance over 560

cycles with a high average CE of 99.32% (Fig. 6a). Remarkably, as increasing the areal capacity to 4 mAh cm$^{-2}$, YP-Cu delivers a high average CE of 99.19% for 250 cycles with negligible attenuation (Fig. 6b). Even when the areal capacity reaches to 6 and 8 mAh cm$^{-2}$, YP-Cu also remains stable for 140 and 100 cycles with high average CEs of 99.21% and 99.05% without distinct oscillation under the lean electrolyte of 11.67 and 8.75 μL mAh$^{-1}$, respectively (Fig. 6c and Supplementary Fig. 30). On the contrary, the CE of bare Cu cells show severe fluctuation within 100 cycles in the same tested conditions (Fig. 6). More importantly, when running the cells with higher current density of 2 mA cm$^{-2}$, YP-Cu still achieve high and stable average CEs above 99.27% and 99.20% for 500 cycles and 200 cycles at plating capacity of 2 mAh cm$^{-2}$ and 4 mAh cm$^{-2}$, respectively. (Fig. 6d and Supplementary Fig. 31).

The corresponding voltage profiles are displayed in Fig. 6e under the areal capacity of 4 mAh cm$^{-2}$ at various cycles, where YP-Cu exhibits flat and smooth voltage plateaus with a low initial overpotential of ~80 mV followed by slightly decrease to 58 mV in the 10th cycle, and maintains low values (50 mV) with almost no change in the subsequent cycles (Fig. 6f), indicating the constant formation of stable interface. Whereas, bare Cu presents fluctuated polarization curves with a larger overpotential of ~106 mV during the first cycle and remains at high values (78 mV) until the cell failed (Fig. 6f and Supplementary Fig. 32). As the current density increases to 2 mA cm$^{-2}$, the overpotential of the YP-Cu electrode still remains stable at different cycles (~ 85 mV). Comparatively, bare Cu shows much higher overpotential of 179 mV in the first cycle, followed by a decrease to 127 mV in the 10th cycle and remains at high values (160 mV) until the cell failed (Supplementary Fig. 33). The high and stable CE of YP-Cu suggest a highly reversible reaction, contributing to superior interface stability.

Symmetrical Li||Li cells were also employed to investigate the Li plating/stripping performance of YP-Li anode. As shown in

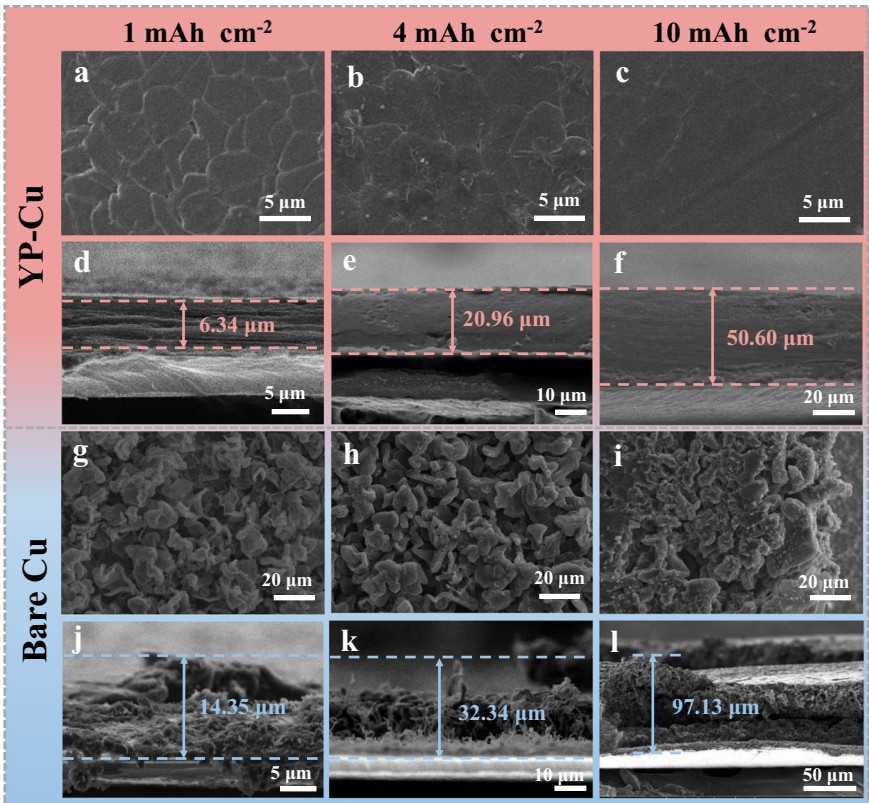

**Fig. 5 | Morphologies of the Li plating on YP-Cu and bare Cu electrodes after the first cycle at different areal capacities. a–c** Top-view and **d–f** cross-sectional SEM images of YP-Cu electrode at the plating process after the first cycle with various capacities of 1, 4, and 10 mAh cm⁻² at 1 mA cm⁻². **g–i** Top-view and **j–l** cross-sectional SEM images of bare Cu electrode at the plating process after the first cycle with various capacities of 1, 4, and 10 mAh cm⁻² at 1 mA cm⁻².

Supplementary Fig. 34, the symmetrical cell with the YP-Li anode delivers Li plating/stripping overpotentials of 23, 37, 58, 76, and 89 mV with a fixed deposition capacity of 1 mAh cm⁻² when the current density increased from 0.25 to 0.5, 1, 2, and 3 mA cm⁻², respectively, which are much lower than that of cell with bare Li anode. Moreover, the YP-Li‖YP-Li cell maintains long-term stability for 1200 h with 2 mAh cm⁻² at 1 mA cm⁻². By comparison, the cell with bare Li anode displays significantly larger overpotential and experiences severe voltage oscillations (Supplementary Fig. 35a). When the current density increases to 2 mA cm⁻², YP-Li anode still demonstrates stable Li plating/stripping overpotential for over 1000 h under 2 mAh cm⁻², whereas bare Li anode shows larger and more pronounced voltage fluctuations even in the initial cycles (Supplementary Fig. 35b). The striking differences in cycling performance strongly demonstrate the superiority of the YP protective layer in promoting dendrite-free Li plating/stripping behavior and maintaining a stable lithium-electrolyte interface.

## Cycling performances of coin/Ah-type full cells under practical conditions

Lastly, to verify the feasibility of YP layer for practical applications, the YP-Li anode was coupled with $LiNi_{0.8}Co_{0.1}Mn_{0.1}O_2$ (NCM811) cathode to assemble full cell in CR2032 coin cells. As shown in Fig. 7a, when high cathode areal capacity of 4.0 mAh cm⁻², lean electrolyte of 10 μL mAh⁻¹ and thin lithium with the thickness of 20 μm are adopted, the YP-Li‖NCM811 cell presents a high capacity retention of 83.78% at 1 C after 200 cycles. In contrast, the bare Li‖NCM811 cell delivers a nearly similar initial capacity under the same test conditions, but suffers a rapid capacity decay in 30 cycles. Additionally, the YP-Li‖NCM811 cell displays stable voltage plateaus with low voltage polarization during cycling, whereas a rapid increase in potential hysteresis is observed for the bare Li‖NCM811 cell (Supplementary Fig. 36).

Based on the effect of YP layer on coin-typed full cells, Ah-typed pouch cell under harsh conditions were also fabricated to further explore the actual practicalities of YP layer for high-specific-energy LMBs. To achieve a specific energy beyond 400 Wh kg⁻¹ at a single-cell level under realistic conditions, the cathode capacity must be above 4.0 mAh cm⁻², and both the Li metal amount and the electrolyte dosage must be tightly restricted (that is, N/P ratio <2 and E/C ratio <3 g Ah⁻¹)[43,44]. The pouch cell in this work comprises a high-areal-capacity NCM811 cathode (6 mAh cm⁻² on each side) and an ultrathin YP-Li foil (100 μm thick, 10 mAh cm⁻² on each side), giving a N/P ratio of only 1.67. Meanwhile, the E/C ratio was limited at 1.98 g Ah⁻¹ (Fig. 7b). Detailed cell parameters, including Al foil, separator, package, and lugs, were presented in Fig. 7c and Supplementary Table 2. Impressively, the assembled YP-Li‖NCM811 pouch cell with high initial capacity of 4.2 Ah delivers a high cell-level energy of 468 Wh kg⁻¹, based on the masses of cell components. The cycling stability of fabricated pouch cell can be further explained in the voltage profiles in different cycles, where the voltage polarization of the YP-Li‖NCM811 pouch cell was almost constant during the repeated charge and discharge process (Fig. 7d). The long-term cycling stability is presented in Fig. 7e, in which the pouch cell is cycled for 150 cycles at 0.5 C (3 mA cm⁻²) under the pressure of 250 kPa with a high capacity retention of 87.42%. More importantly, the high energy density (468 Wh kg⁻¹) and excellent stability (0.08% decay/cycle) of our designed YP-Li‖NCM811 pouch cell outperform most of reported Li metal pouch cells (Fig. 7f and Supplementary Fig. 37), especially under such extremely harsh testing conditions[16,19,45–58]. Moreover, if the E/C ration is further reduced to 1.4 Ah g⁻¹, the energy density can reach 504.58 Wh kg⁻¹. The above findings elucidate that YP-protected Li metal anode possesses outstanding cycling stability even under high areal capacity, demonstrating great prospects for the real-life applications of high-energy-density and long-lifespan LMBs.

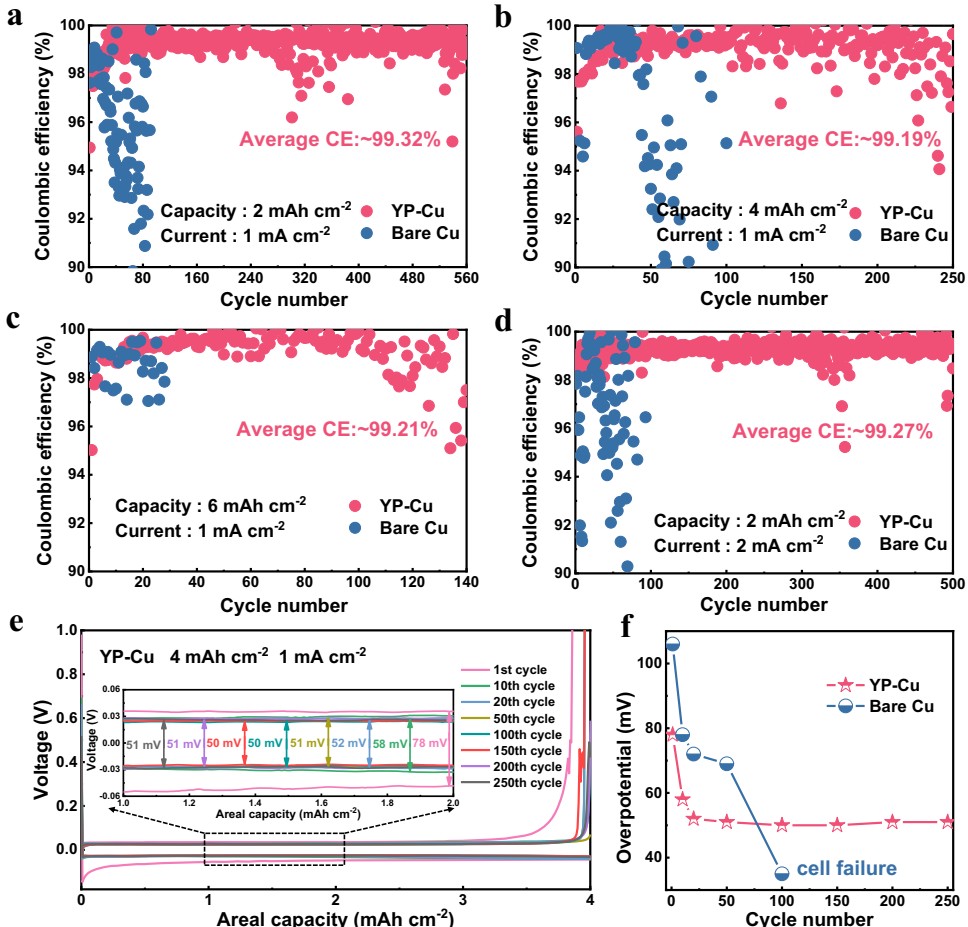

**Fig. 6 | Electrochemical performance of Li||Cu asymmetric cells.** Coulombic efficiency of Li||YP-Cu and Li||Cu asymmetric cells with various capacity of **a** 2, **b** 4, and **c** 6 mAh cm⁻² at 1 mA cm⁻². **d** Coulombic efficiency with 2 mAh cm⁻² at 2 mA cm⁻². **e** voltage profiles of Li||YP-Cu asymmetric cell with 4 mAh cm⁻² at 1 mA cm⁻². **f** Summary of the overpotential value as a function of cycle number.

## Discussion

A stable high-areal-capacity lithium metal anode is developed through a synergetic regulation of SEI mechanics and crystallographic orientation of Li to realize the 460 Wh kg⁻¹ LMBs. Specifically, the crystal's surface energy is changed and Y-doped (200) crystal plane has the lowest surface energy, even lower than the (110) plane, which promotes the transformation of the preferred oriented growth to (200) crystal plane from (110) plane and relieves the reactivity of lithium metal with the electrolyte, thus endowing a dense lithium deposition. Furthermore, a LiF-rich SEI can also be generated upon cycling, achieving rapid Li-ion transport. Benefiting from the suitable modulus of YP layer, mechanical stability of SEI can be enhanced, which can tolerate the huge volume change during cycling. Noticeably, the assembled Li||YP-Cu asymmetric cells provide a high average CE of 99.21% for 140 cycles even under high areal capacity of 6 mAh cm⁻² with lean electrolyte of 11.67 μL mAh⁻¹. More profoundly, when coupled with high-areal-capacity NCM811 cathode of 6 mAh cm⁻², the YP-Li||NCM811 pouch cell exhibits high energy density of 468 Wh kg⁻¹ and exceptional cycling stability of 0.08% decay per cycle under rigorous conditions.

## Methods
### Materials
Yttrium fluoride (YF₃) was purchased from Macklin Biochemical Co., Ltd (Shanghai, China). Polymethyl methacrylate (PMMA, Mₙ = 35,000), tetrahydrofuran (THF), and N-methyl-2-pyrrolidone (NMP) were purchased from Sinopharm Chemical Reagent Co., Ltd in analytical purity.

Polyvinylidene difluoride (PVDF, Arkema) and Super P carbon black (TIMCAL) were used as-purchased. Commercial LiNi₀.₈Co₀.₁Mn₀.₁O₂ (NCM811) was used as the cathode.

### Preparation of YP-Cu/YP-Li electrodes
The mixture (YP) of YF₃ particles and PMMA with a weight ratio of 1:1 was added in tetrahydrofuran (THF) with a total concentration of 0.1 wt.%. Subsequently, the uniform slurry was evenly coated onto Cu or Li foil via blade casting. Then, the coated samples underwent vacuum drying at 60 °C for 5 h to ensure the solvent totally evaporation.

### Assembly of the 4.0 Ah lithium metal pouch cells
Pouch cells were assembled in a dry room with a dew point below −56 °C. The fabrication process involved ultrasonic welding, packaging, and electrolyte injection, followed by vacuum sealing. NCM811 was used as the cathode material. The slurry, composed of NCM811, PVDF binder, conductive additive Super P, and conductive additive VGCF in a weight ratio of 95.5: 2.2: 1.8: 0.5 in NMP, was uniformly coated onto both sides of 12 μm-thick Al foil and dried in a vacuum oven at 80 °C for 24 h. The obtained electrodes were punched into rectangular pieces (56 × 80 mm²). In our work, the single-sided mass loading of NCM811 was ~30 mg cm⁻², corresponding to ~6 mAh cm⁻² on each side. Li foil coated with YP layer with a thickness of ~100 μm was employed as the anode, providing an areal capacity of 10 mAh cm⁻² for a single side. The N/P ratio of the YP-Li||NCM811 pouch cell was 1.67. The electrolyte consisted of LiDFOB (0.6 M) and LiBF₄ (0.6 M) in fluoroethylene carbonate (FEC) and diethyl carbonate (DEC) (v/v = 1:2).

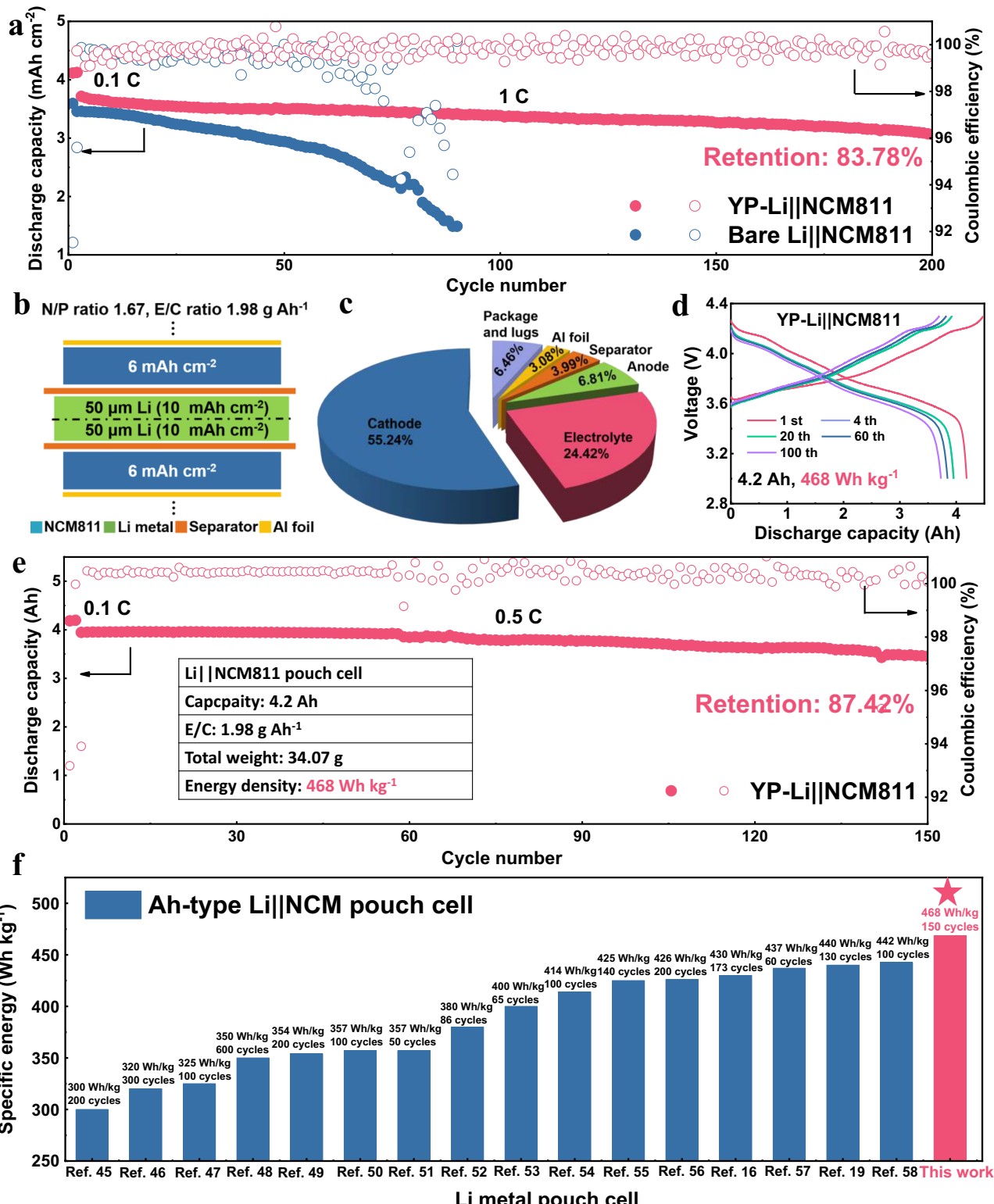

**Fig. 7 | Electrochemical properties of the coin-type and Ah-type full cells.**
**a** Long-term cycling performance of the coin-type full cells with and without YP at
1 C. **b** A schematic illustration of our fabricated 4.2 Ah pouch cell under practical
conditions. **c** The pie chart of the weight distribution of all cell components in a
pouch cell. The energy density in this work is calculated by the whole weight
including the package and lugs. **d** Typical charge/discharge curves of YP-Li||
NCM811 pouch cell at different cycles. **e** Cycling performance of the pouch cell
under practical conditions. **f** The performance of high-energy-density (over
300 Wh kg⁻¹) Ah-type Li metal pouch cells in published literatures and this work.

Notably, the electrolyte amount to capacity ratio (E/C ratio) was limited at 1.98 g Ah$^{-1}$. Celgard 2325 membrane (25 μm) was used as the separator. Further detailed parameters are presented in Supplementary Table 2. Pouch cells underwent cycling within the voltage range of 3.0–4.3 V under 25 °C, initially at 0.1 C (1 C = 6 mA cm$^{-2}$) for the first two formation cycles, followed by 0.1 C charge and 0.5 C discharge for the subsequent cycles under the same voltage range. Cycling performance was evaluated at the pressure of 250 kPa. It was noted that the capacity retention of pouch cells was calculated from the third cycle as the starting capacity. The energy density was calculated by the following equation:

$$E = \frac{C \times V}{M} \tag{1}$$

Where C is the cell capacity (Ah), V is the Mid-value voltage (V), and M is the total mass of the cell (kg) including anode, cathode, electrolyte, separator, package, and lugs.

## Electrochemical measurements

To explore the electrochemical performance of the asymmetric Li||Cu half cells, CR2032 coin-type cells were assembled in an argon-filled glovebox (H$_2$O < 0.1 ppm, O$_2$ < 0.1 ppm) with 70 μL electrolyte, which contains 4 M bis(fluorosulfonyl)imide lithium (LiFSI) in 1,2-dimethoxyethane (DME). Celgard 2325 membrane was used as the separator. The YP-Cu or bare Cu foil were used as working electrodes and Li foil as counter electrodes. The electrochemical performance of as-prepared Li||Cu cells were evaluated at the current density of 1 or 2 mA cm$^{-2}$ with various capacities from 2 mAh cm$^{-2}$ to 4, 6, and 8 mAh cm$^{-2}$ conditions. For the Li||NCM811 coin-type full cells, the thickness of the Li foil was 20 μm, the cathode mass loading was ~20.6 mg cm$^{-2}$ (~4 mAh cm$^{-2}$). The carbonate-based electrolyte (0.6 M LiDFOB and 0.6 M LiBF$_4$ in 1:2 FEC/DEC) was employed to tolerate the high-voltage of NCM811. The Li||NCM811 coin cells were measured using a Neware battery system (Shenzhen, China) within the voltage range of 3.0–4.3 V at 25 °C. Moreover, the capacity of fourth cycle was used as the initial capacity to calculate the capacity retention. The CHI electrochemical workstation was conducted to test corrosion current density after different cycles at a scanning rate of 1.0 mV s$^{-1}$ within the voltage range of −0.12–0.12 V. Solartron Energy lab XM was utilized to investigate the electrochemical impedance spectroscopy (EIS) after different cycles with a frequency ranging from 1 Hz to 100 kHz at an amplitude of 5 mV. Activation energy measurements was tested through EIS characterization in an incubator under different temperatures from 25 to 75 °C with 10 °C separation. Activation energy value was calculated through the equation

$$\frac{T}{R_{SEI}} = A \exp\left(\frac{-E_a}{RT}\right) \tag{2}$$

where $E_a$ is the activation energy, T is the absolute temperature, R is the gas constant, $R_{SEI}$ is the interfacial Li$^+$ transfer resistance, and A is the pre-exponential factor.

## Characterization

In situ X-ray diffraction (XRD) was recorded using Aeris within the range of 30–70°. The Li||Li symmetric cell was rested for 4 hours, and then it was cycled under a high areal capacity of 4 mAh cm$^{-2}$ with a current density of 1 mA cm$^{-2}$ on a LAND multichannel battery tester (CT2001A) with the simultaneous collection of XRD data upon Li plating. These cells, sealed in the cell molds containing Be, were assembled in an argon-filled glovebox (H$_2$O < 0.1 ppm, O$_2$ < 0.1 ppm). Ex situ XRD was conducted using Aeris to identify the crystal structure evolution of the samples after different cycles. Before the

XRD test, the samples were sealed with polyimide tapes in an argon-filled glove box to isolate from oxygen and moisture in the air. The morphologies of Li plating/stripping and elemental distributions were characterized by the field-emission scanning electron microscopy (SEM) acquired on FEI Verios 460. The SEI components on Li metal anodes were performed by X-ray photoelectron spectroscopy (XPS) on an ESCALAB Xi+ spectrometer (Thermo Fisher) using Al Kα X-ray radiation. All the samples were transferred to the holder to avoid exposure to air. The Cryo-TEM characterization was performed on Filed Emission Transmission Electron Microscope (FE-TEM, Talos-F) operated at 200 kV to analyze the composition and structure of SEI. ToF-SIMS (M6) also investigated SEI components. AFM (NT-MDT) was used to analysis the Young's modulus. Raman spectra was performed under 785 nm laser excitation (Thermo Scientific DXR3xi).

## Simulations

Based on the density functional theory, first-principles calculations were performed using the projected augmented plane-wave method implemented in the Vienna Ab-initio Simulation Package (VASP, 6.1.0). Perdew-Burke-Ernzerhof (PBE) functional of generalized gradient approximation was used to treat electron exchange and approximation, in which the energy cutoff was chosen to be 520 eV. The surface energy of bare Li and Y-doped Li is defined as $\bar{E}_{s-Li} = (E_S - E_{b-Li})/2S$ and $\bar{E}_{s-Li_jY_j} = (E_S - E_{b-Li-Y})/S - \bar{E}_{s-Li}$, respectively, where $E_S$ is total energy of slab, $E_{b-Li}$ and $E_{b-Li-Y}$ are the energies of bulk Li and Li$_{95}$Y for 6 atom layers, and the $S$ is the surface area. Bulk body-centered cubic (BCC) Li, bare Li slabs and Y-doped Li supercell slabs were sampled in k space of 11 × 11 × 11, 5 × 5 × 1 and 5 × 5 × 1 grids in the Brillouin zone, respectively. Only one Y atom was doped to the surface, making a Li/Y ratio of 1/95 per slab. Atomic positions on the surface of the slabs were completely relaxed, but the interior atomic layers were fixed in the slabs. Based on relaxed bulk structure of Li and Y, six atomic layer slabs perpendicular to different planes, (200), (110), and (211) plane, were built and fully relaxed. The positions of all atoms were completely relaxed until the energy between two consecutive steps is less than 10$^{-5}$ eV, and the maximum force on each atom was 0.01 eV/Å. The periodic boundary conditions were applied along the x and y axis, and vacuum space was applied at least 20 Å along the z axis to securely avoid artificial interactions between the slabs.

The finite element method (FEM) simulations were conducted to investigate the stress distribution. The 50 nm × 2000 nm PMMA layer locates above SEI with the size of 50 nm × 20 nm. The detailed material properties and parameters were listed in Supplementary Table 3.

Nernst-Planck equation was employed to describe Li$^+$ transportation in electrolyte, PMMA and SEI

$$J = -D\nabla c - z\frac{D}{RT}Fc\nabla\varphi \tag{3}$$

where J represents Li$^+$ flux, D is the diffusion coefficient, c is the concentration, z is the charge number, F is the Faraday's constant, φ is the potential.

The von mises stress was solved by mechanical equilibrium equation

$$\nabla \cdot \sigma = 0 \tag{4}$$

where σ donates Cauchy stress tensor.

Little or enriched distributed LiF were added into SEI. Y metal was also doped into Li metal described by uniform distributed noise. The interface of Li and SEI was set as a prescribed displacement boundary condition. Both sides of the electrolyte and SEI were set periodic boundary conditions. The cathode was applied with a current density of 10 A m$^{-2}$, and the anode was grounded.

**Reporting summary**

Further information on research design is available in the Nature Portfolio Reporting Summary linked to this article.

## Data availability

The authors declare that all data supporting the finding of this study are available within the paper and its supplementary information files. All raw data generated during the current study are available from the corresponding author upon request.

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

## Acknowledgements

This work was supported by the National Natural Science Foundation of China (No. 22279100 (J.S.) and 21875181 (J.S.)) and Natural Science Basic Research Program of Shaanxi (No. 2019JLP-13 (J.S.)). The authors would like to thank Chenyu Liang, Yan Liang, and Wei Wang at Xi'an Jiaotong University for their help in materials characterization.

## Author contributions

J.S. supervised the research. J.S. and Y.Z. conceived the idea and designed the experiments. Y.Z. prepared materials, carried out the experimental planning, electrochemical measurements, characterization, and data analysis. Y.Z., Q.N., Y.L., and J.X. assembled the pouch cells and discussed the electrochemical tests. R.Q. and P.Z. contributed to experimental design. J.D., Y.H., and S.C. conducted the simulations and calculations. C.L., B.S., H.F., and F.L. assisted with the data analysis. J.S. and Y.Z. wrote the manuscript. All authors commented on the manuscript.

## Competing interests

The authors declare no competing interests.
