## [Peer Review File · Nature Communications]

Synergetic Regulation of SEI Mechanics and Crystallographic Orientation for Stable Lithium Metal Pouch CellsREVIEWER COMMENTS

Reviewer #1 (Remarks to the Author):

This work uses YF3/PMMA coating to treat the Cu substrate or the Li anode, leading to a 4Ah, >400 Wh/kg pouch cell that can go through over 150 cycles. The lithium metal battery performance is encouraging and comparable to the state-of-the-art. But the scientific interpretation and understanding of why such approach works is at most ambiguous, if not inaccurate. Authors need to work on the science part to make the paper acceptable to Nature Communications. Detailed comments are as follows:

1. Authors need to explain why they used yttrium compound to start with. Is this a lucky success out of trial-and-error (which is also acceptable) or yttrium compounds have some particular properties that can lead to good SEI? similarly, why PMMA? the justifications of using yttrium compound and PMMA should be discussed.
2. The claim of Y doping into Li is dubious and weakly supported. Authors must clarify this point because Y doping into Li is the basis of a lot of other arguments, for example, the Y-Li surface energy calculation. Authors refer to Figure S6 (selected area electron diffraction) and Figure S7 (EDS mapping) as the experimental evidence of Y-doped Li. For SAED, there is no discussion of why it could be used for the evidence. If the authors imply that the conclusion was drawn from the lattice parameter evaluation, then this is not a convincing argument. It is known that electron diffraction cannot give accurate lattice parameters. For EDS mapping, what is shown in Figure S7 can have multiple interpretations. YF3 with uniform spatial distribution can well explain Figure S7. The result tells nothing about Y doping in Li. As a matter of fact, doping Y into Li may be difficult as Y has hexagonal structure but Li is cubic.
3. The electrolyte used in this work is based on carbonates. Carbonates can have better kinetics than LHCE and fluorinated ether and therefore better kinetics. But it is known that carbonate is not stable against lithium metal and it is not surprising to see the Coulombic efficiency in Li||Cu cell is at most 99.3%. This is lower than those reported in the literature (for example, LHCE and fluorinated ether can achieve around 99.5%). The potential instability of electrolyte against lithium metal may be an increasingly serious concern as the cycle goes on. What is the pressure used for coin cell and pouch cell cycling? pressure is a very important parameter for lithium metal batteries. It can greatly influence the cycle life. authors should provide this information in their table of pouch cell parameters.
4. In Figure 1a, why the (110) peak shifts but (200) peak does not change during lithium plating? In Figure S8, why the (110) peak appears very sharp but the (200) peak is so broad? they are supposed to have similar peak width. It seems that Be window produces some peaks around the Li (200) peak. Would this undermine the basis of in situ XRD data analysis?
5. There is no description of how in situ XRD was carried out. This is an important experiment because authors mainly use its result to support the claim of (200) lithium deposition.
6. Authors claim that YP coating leads to 4 times less side reaction. How the magnitude of side reaction is measured? It can not be simply indicated by the cycle life which is the overall outcome determined by side reactions, impedance, and many other factors.

Reviewer #2 (Remarks to the Author):

In this manuscript, the authors report a synergetic regulation strategy of SEI mechanics and crystallographic orientation of Li via a YF₃/polymethyl methacrylate composite (YP) skin. Among them, a Y-doped Li metal can be generated by the reaction of YP with Li metal, and Y-doped (200) plane has the lowest surface energy, which enables the preferred oriented growth of Li metal along the (200) crystal plane. Thus, ultimately accomplishing dense and smooth deposition even at high-areal capacity of 10 mAh cm⁻². The authors also studied the SEI systematically. At last, a 4.2 Ah Li metal pouch cell reaches a high energy density of 468 Wh kg⁻¹ and undergoes 150 cycles. This work is interesting. However, the following comments have to be addressed before publication.

1. For the figure of the performance of high-energy-density (over 300 Wh kg⁻¹) Ah-type Li metal pouch cells, some recently published references are missing.
2. The characterization of Li metal in pouch cell should also be provided to tell the effect of the coating layers in practical cells.
3. Symmetric cells with different current density should also be provided to tell the performance.
4. How could the authors ensure the uniform distribution of the YP in PMMA? Is it small particles, clusters or molecules?
5. "Therefore, it can be concluded that Y-doped Li metal can reduce the reactivity of Li metal with electrolytes, thus enabling a stable lithium-electrolyte interface." How can the author make sure that this effect comes from the T doping rather than the protection of PMMA?
6. "These changes effectively passivate Li metal, thereby significantly reducing undesired side reactions between Li and electrolytes by 4 times." Again, how can the author make sure that this effect comes from the T doping rather than the protection of PMMA?
7. There are some minor errors in the manuscript, such as page 12 figure 2e, SI page 9 figure S8...

Reviewer #3 (Remarks to the Author):

These authors report on a Yttrium fluoride-polymer composite that appears to form favorable SEI on the Li metal anode. Authors found that the YF₃ reacts with Li metal to form LiF and a Yttrium-doped Li material that alter the surface energetics and crystallographic growth features for deposited Li. On these basics authors claim to have created an energy dense Li cell that according to the paper title delivers 460 Wh/kg.

I have a number of concerns that make it less clear about the importance of the study for Nature Communications.

1. The concept of using a particular/special additive in electrolyte or at interface to improve cycling of Li is not new or novel. There are many papers of this type in the literature and few have so far actually led to advances. What would be of interest to the readership is the underlying principle by which additive works and how it can be extended to other additives.
2. Why Yttrium is used in first place? YF₃ is expensive (more than 2000 USD/kg), which would make battery that depend on it impractical. If authors are correct that the reason for

the high capacity and long life is that the Y-doped Li favors the (200) facet, same approach might be adopted to another less so expensive chemistry. It is a serious weakness that authors make no effort to show this.

3. There are already previous studies that show that SEI composed of F-rich species enhance cycle life of the Li anode. Authors claims that a different mechanism is responsible for results reported but this conclusion is not strong because for every mole of YF₃ 3 moles of LiF are made. The special role claimed for the Y-doped Li would be more convincing if authors showed that Yttrium salts that do not contain F achieve the similar results.

4. The DFT analysis assume that the electrolyte does play a role. This is likely untrue for a reactive metal like Li where SEI formation will lower the effectiveness of a templating coating as the Li deposit thickness increases. Said differently, if the Y-doping can indeed control the crystal mode for Li in the first layer, is it really possible to completely passivate the subsequent Li deposit from reaction with electrolyte? Also, is it really possible to achieve dominantly 200 growth at 100 μm Li thicknesses as claimed. To support authors claims we would need to see current vs time data for the corrosion reaction measured at different Li thickness.

5. Is specific energy value stated in the title and reported in Figure 6 theoretical values? I see that the 468 Wh/kg value in Figure 6 is actually based on the first cycle capacity during formation (C/10) cycling. The value is lower by the 4th cycle and I estimate a cell specific energy close to that in Ref 51. At the very least, authors will need to fully explain how specific energy values given in Figure 6 are determined and under what conditions the new YP-LI result are superior to ref 51.

Choice of PMMA polymer is not clear? Could other polymer work just as well? If not, explain this.

Response to Reviewer #1

Reviewer #1:

This work uses YF₃/PMMA coating to treat the Cu substrate or the Li anode, leading to a 4Ah, >400 Wh/kg pouch cell that can go through over 150 cycles. The lithium metal battery performance is encouraging and comparable to the state-of-the-art. But the scientific interpretation and understanding of why such approach works is at most ambiguous, if not inaccurate. Authors need to work on the science part to make the paper acceptable to Nature Communications. Detailed comments are as follows:

Response:

We would like to express our sincere appreciation for your careful reading and valuable comments. In this work, we highlight the significance of a YF₃/PMMA (YP) composite skin, aiming to establish the mechanical stability of the solid electrolyte interphase (SEI) and the dense deposition of Li metal. This is achieved through a synergistic regulation of SEI mechanics and the crystallographic orientation of the Li metal anode.

In response to your insightful suggestions, we have delved deeper into the fundamental understanding of the working mechanism behind the YF₃/PMMA (YP) composite skin. Additionally, we have provided a comprehensive explanation for our choice of YF₃ and PMMA, as detailed in *Reviewer Comments 1* and *2*. To fortify our conclusions, we have incorporated new experimental evidence, including XPS spectra and XRD patterns before and after Li plating, in the revised manuscript. These changes aim to enhance the robustness of our findings and address the concerns raised during the review process.

Reviewer Comment 1: *Authors need to explain why they used yttrium compound to start with. Is this a lucky success out of trial-and-error (which is also acceptable) or yttrium compounds have some particular properties that can lead to good SEI? similarly, why PMMA? the justifications of using yttrium compound and PMMA should be discussed.*

Response:

We are thankful to the reviewer's comment. To achieve a highly stable Li metal anode under high deposition capacity conditions, addressing the challenges of uncontrollable dendrite growth and unstable Li-electrolyte interface is essential. Fundamentally, the formation of Li dendrites undergoes two steps involving nucleation and growth of the Li crystal.¹ As with other metal anodes, Li metal grows continuously on its early-formed nuclei during cycling. The final deposition morphology tightly relies on the preferential crystallographic orientation of Li metal crystal.^{2,3}

Intrinsically, (110), (200), and (211) crystal planes are normally the dominant crystallographic features in the Li metal. Among them, the (110) crystal plane is the most densely packed and thus has the lowest-energy surface, which is regarded as the preferential growth crystal plane of Li metal anode.⁴ Thermodynamically, Li dendrites will eventually be formed if the Li metal keeps growing along the (110) crystal plane.^{5,6} Therefore, regulating Li metal's preferential growth orientation is necessary for fundamentally resolving the issues of Li dendrite growth and unstable interfaces.

The preferred crystallographic orientation depends on the surface energy of the crystal.^{7,8} Doping with rare-earth element is considered as an efficient strategy to change the surface energy and consequently regulate the preferred crystallographic orientation.^{9,10} Theoretical studies have demonstrated that Li metal with low surface energy result in high interfacial energy, thereby promoting the horizontal migration of Li^+ and effectively suppressing the vertical growth of Li metal. In addition, the lower surface energy contributes to the formation of larger Li deposit particles, which is beneficial for alleviating the reactivity between Li metal and electrolytes.^{11,12} Considering yttrium's favorable affinity with Li, along with its large atomic radius,¹³ we hypothesized that doping Y into Li metal could tune and reduce the surface energy of Li metal, and thus regulate the preferred orientations, ultimately increasing the nucleation radius of Li metal. This fundamentally further reduces the undesired side reactions between Li and electrolytes and inhibits Li dendrite growth.

Meanwhile, LiF has high interfacial energy, high chemical stability, and a low Li^+

diffusion barrier.^{14, 15} Therefore, LiF is believed to suppress lithium dendrite growth and boost the cycle life of lithium metal batteries. Constructing a stable and LiF-rich SEI on Li metal anodes is necessary for practical lithium metal batteries.¹⁶ More importantly, YF₃ can react with deposited Li to form Y and LiF. The formed LiF-rich SEI leads to rapid charge transfer kinetics. Therefore, we chose YF₃ as a model to achieve a Y doping into Li metal and a stable LiF-rich SEI. This is expected to fundamentally tune the preferred orientation of Li metal and reduce the side reactions between Li metal and electrolytes, contributing to the realization of high-energy-density lithium metal batteries. Additionally, other fluorides with similar properties may also exhibit analogous effects. Further experiments and systematic explorations are currently underway to investigate these possibilities.

Furthermore, PMMA with excellent polymeric segmental motion owns a high ionic conductivity and elastic modulus. The carbonyl of PMMA has strong interactions with Li⁺, which could regulate uniform Li deposition.^{17, 18} Therefore, PMMA can serve as a protective sheath to enhance mechanical stability and uniform stress distribution of SEI, thus preventing its structure from being destroyed upon cycling.

Based on the aforementioned hypothesis, we selected a YF₃/PMMA (YP) composite skin as a representative sample to realize mechanical stability of SEI and dense deposition of Li metal through synergetic regulation of SEI mechanics and crystallographic orientation of Li metal anode. Specifically, Y-doped (200) plane has the lowest surface energy, which enables the preferred oriented growth of Li metal along the (200) crystal plane rather than conventional (110) plane and relieves the reactivity between Li metal and electrolytes during cycling, ultimately accomplishing dense and smooth deposition even at high-areal-capacity of 10 mAh cm⁻². Moreover, the formed LiF-rich SEI leads to rapid charge transfer kinetics. Meanwhile, YP layer with a suitable modulus of ~1.02 GPa can enhance mechanical stability and uniform stress distribution of SEI, thus preventing its structure from being destroyed upon cycling. Therefore, a high-areal-capacity Li metal anodes with dense deposition and mechanical stability have been developed for high-specific-energy lithium metal batteries. As a result, a 4.2 Ah YP-Li||NCM811 pouch cell reaches a high energy density

of 468 Wh kg⁻¹ and an excellent capacity stability of 0.08% decay/cycle under realistic conditions. The detailed discussion has been added in the revised manuscript (page 7, line 1-11; page 8, line 1-14) and is also shown below.

“Achieving dense deposition at the bottom and mechanical stability at the top are critical for high-areal-capacity Li metal anodes to meet high-energy-density lithium metal batteries. At this point, we aimed to construct a composite protective skin on Li metal anodes that can relieve the reactivity between Li metal and electrolytes in the bottom layer and enhance mechanical stability of SEI in the top layer upon cycling. Therefore, regulating Li metal’s preferential growth orientation and building a stable SEI on Li metal anode are necessary for realizing high-energy-density lithium metal batteries. The preferred crystallographic orientation depends on the surface energy of the crystal.^{7, 8} Doping with rare-earth element is considered as an efficient strategy to change the surface energy and consequently regulate the preferred crystallographic orientation.⁹ The *in-situ* reaction between YF₃ and Li metal results in the generation of Y-doped Li metal and LiF. Y doping into Li metal could tune and reduce the surface energy of Li metal, and thus regulate the preferred orientations, ultimately reducing undesired side reactions between Li metal and electrolytes and inhibiting Li dendrite growth. Meanwhile, the formed LiF-rich SEI could accelerate charge transfer kinetics. Furthermore, polymethyl methacrylate (PMMA), with excellent polymeric segmental motion, high ionic conductivity and elastic modulus^{17, 18}, can enhance mechanical stability and uniform stress distribution of SEI, thus preventing its structure from being destroyed upon cycling and contributing to the formation of a stable SEI.

Based on the aforementioned hypothesis, we propose a synergetic regulation strategy of SEI mechanics and crystallographic orientation of Li through a YF₃/polymethyl methacrylate (PMMA) composite skin (denoted as YP).”

References:

1. Pei, A., et al. Nanoscale nucleation and growth of electrodeposited lithium metal. *Nano Lett.* **17**, 1132-1139 (2017).
2. Chen, X.-R., et al. Review on Li deposition in working batteries: From nucleation to early growth. *Adv. Mater.* **33**, 2004128 (2021).
3. Su, X., et al. Mechanisms for lithium nucleation and dendrite growth in selected carbon

- allotropes. *Chem. Mater.* **29**, 6205-6213 (2017).
- Vitos, L., et al. The surface energy of metals. *Surf. Sci.* **411**, 186-202 (1998).
 - Hagopian, A., et al. Thermodynamic origin of dendrite growth in metal anode batteries. *Energy Environ. Sci.* **13**, 5186-5197 (2020).
 - Gao, X., et al. Thermodynamic understanding of Li-dendrite formation. *Joule* **4**, 1864-1879 (2020).
 - Cao, Y., et al. Solvothermal synthesis of TiO₂ nanocrystals with {001} facets using titanate nanobelts for superior photocatalytic activity. *Appl. Surf. Sci.* **391**, 311-317 (2017).
 - Connell, J.-G., et al. Crystal orientation-dependent reactivity of oxide surfaces in contact with lithium metal. *ACS Appl. Mater. Interfaces* **10**, 17471-17479 (2018).
 - Chen, T., et al. Manipulating the crystal plane angle within the primary particle arrangement for the radial ordered structure in a Ni-rich cathode. *Chem. Sci.* **14**, 13924-13933 (2023).
 - Kim, Y., et al. Insights into the microstructural engineering of cobalt-free, high-nickel cathodes based on surface energy for lithium-ion batteries. *Adv. Energy Mater.* **13**, 2204054 (2023).
 - Lopez, J., et al. Effects of polymer coatings on electrodeposited lithium metal. *J. Am. Chem. Soc.* **140**, 11735-11744 (2018).
 - Zhou, B., et al. Modification of Cu current collectors for lithium metal batteries – A review. *Prog. Mater. Sci.* **130**, 100996 (2022).
 - Wang, S.-H., et al. Tuning wettability of molten lithium via a chemical strategy for lithium metal anodes. *Nat. Commun.* **10**, 4930 (2019).
 - Liu, Y., et al. Self-assembled monolayers direct a LiF-rich interphase toward long-life lithium metal batteries. *Science* **375**, 739-745 (2022).
 - Chen, M., et al. Marrying ester group with lithium salt: Cellulose-acetate-enabled LiF-enriched interface for stable lithium metal anodes. *Adv. Funct. Mater.* **31**, 2102228 (2021).
 - Yu, Z., et al. Design principles of artificial solid electrolyte interphases for lithium-metal anodes. *Cell Rep. Phys. Sci.* **1**, 100119 (2020).
 - Zhou, J., et al. A new high ionic conductive gel polymer electrolyte enables highly stable quasi-solid-state lithium sulfur battery. *Energy Storage Mater.* **22**, 256-264 (2019).
 - Zhou, Z., et al. A robust, highly stretchable ion-conductive skin for stable lithium metal batteries. *Chem. Eng. J.* **396**, 125254 (2020).

Reviewer Comment 2: *The claim of Y doping into Li is dubious and weakly supported. Authors must clarify this point because Y doping into Li is the basis of a lot of other arguments, for example, the Y-Li surface energy calculation. Authors refer to Figure S6 (selected area electron diffraction) and Figure S7 (EDS mapping) as the experimental evidence of Y-doped Li. For SAED, there is no discussion of why it could be used for the evidence. If the authors imply that the conclusion was drawn from the lattice parameter evaluation, then this is not a convincing argument. It is known that electron diffraction cannot give accurate lattice parameters. For EDS mapping, what is shown*

in Figure S7 can have multiple interpretations. YF₃ with uniform spatial distribution can well explain Figure S7. The result tells nothing about Y doping in Li. As a matter of fact, doping Y into Li may be difficult as Y has hexagonal structure but Li is cubic.

Response:

In response to your concerns, we supplemented XPS spectra and XRD patterns before Li plating and after Li plating, and detailed explanations to demonstrate the Y doping into Li metal in the revised manuscript (page 9, line 1-9, line 13-22; page 10, line 1).

Firstly, we conducted X-ray photoelectron spectroscopy (XPS) spectra of Y 3d_{5/2} and Li 1s for YP-Cu electrode before and after Li plating to investigate the reduction reaction between YF₃ and deposited Li. Before Li plating, the binding energy at ~159.8 eV in the Y 3d_{5/2} spectra is assigned to YF₃ (Supplementary Fig. 5b). After Li plating, the binding energy at ~157.8 eV in the Y 3d_{5/2} spectra (Supplementary Fig. 5e) and 55.8 eV in the Li 1s spectra (Supplementary Fig. 5f) are assigned to metallic Y and LiF,¹⁹ respectively, confirming the reduction of YF₃ to form Y and LiF.

Moreover, Cryo-transmission electron microscopy (Cryo-TEM) was also employed to further confirm the Y doping into Li metal. As depicted in Supplementary Fig. 6, the nanostructures of the deposited Li on the YP-Cu are predominantly composed of Li crystals and LiF crystals. Specifically, the lattice spacing of 0.176 nm corresponds to (200) plane of Li metal, and the lattice spacing of 0.142 nm is assigned to (220) plane of LiF. Importantly, no lattice fringes of doped Y are observed, suggesting that Y doping has not altered the crystal structure of Li metal. Additionally, the Li deposits on the YP-Cu electrode and the corresponding elemental distributions of Y and F elements were also analyzed to further validate the uniform distribution of Y element on the surface of the YP-Cu (Supplementary Fig. 7). These findings demonstrate the uniform distribution of the Y element over the surface of the YP-Cu electrode, confirming that Y has been successfully doped into the Li metal.²⁰

At last, XRD patterns of bare Li and YP-Li before Li plating and during Li plating at 1 mA cm⁻² with increased deposition capacity from 1 to 10 mAh cm⁻² are shown in Fig. R1. All XRD patterns of both bare Li and YP-Li before and after Li plating can match

the crystal structure of Li metal, indicating that Y doping does not significantly alter the crystal structure of Li metal.^{20, 21} These findings are consistent with the observations from Cryo-TEM images.

Supplementary Fig. 5. XPS spectra of C 1s, Y 3d_{5/2}, and Li 1s for the YP-Cu electrode (a-c) before and (d-f) after Li plating.

Supplementary Fig. 6. Cryo-TEM images of the deposited Li on the YP-Cu electrode and corresponding fast Fourier transform images.

Supplementary Fig. 7. Cryo-TEM images of the deposited Li on the YP-Cu electrode and corresponding elemental distributions of Y and F.

Fig. R1. XRD patterns of the (a) bare Li and (b) YP-Li before Li plating and during Li plating at 1 mA cm⁻² with increased deposition capacity from 1 to 10 mAh cm⁻².

References:

19. Xie, Y., et al. Surface modification using heptafluorobutyric acid to produce highly stable Li metal anodes. *Nat. Commun.* **14**, 2883 (2023).
20. Ou, X., et al. Enabling high energy lithium metal batteries via single-crystal Ni-rich cathode material co-doping strategy. *Nat. Commun.* **13**, 2319 (2022).
21. Huang, S., et al. Y-doped Li₈ZrO₆: A Li-ion battery cathode material with high capacity. *J. Am. Chem. Soc.* **137**, 10992-11003 (2015).

Reviewer Comment 3: *The electrolyte used in this work is based on carbonates. Carbonates can have better kinetics than LHCE and fluorinated ether and therefore better kinetics. But it is known that carbonate is not stable against lithium metal and it is not surprising to see the Coulombic efficiency in Li||Cu cell is at most 99.3%. This is lower than those reported in the literature (for example, LHCE and fluorinated ether can achieve around 99.5%). The potential instability of electrolyte against lithium metal may be an increasingly serious concern as the cycle goes on. What is the pressured used for coin cell and pouch cell cycling? pressure is a very important parameter for lithium metal batteries. It can greatly influence the cycle life. authors should provide this information in their table of pouch cell parameters.*

Response:

We are grateful with the reviewer's comment. Actually, the ether-based electrolytes instead of carbonate-based electrolyte is used in our manuscript in the asymmetric Li||Cu cells. As a result, at a current density of 1 mA cm⁻² with the plating capacity of 2 mAh cm⁻², Li||YP-Cu cells display a stable cycling performance over 550 cycles with a high average CE of 99.32%. Remarkably, as increasing the areal capacity to 4 mAh cm⁻²

², Li||YP-Cu cells deliver a high average CE of 99.19% for 250 cycles with negligible attenuation. Even when the areal capacity increases to 6 and 8 mAh cm⁻², the Li||YP-Cu cells still remains stable for 140 and 100 cycles with high average CEs of 99.21% and 99.05% under the lean electrolyte of 11.67 and 8.75 μL mAh⁻¹, respectively. The high CE and excellent stability of YP-Cu **under high areal capacity** outperform most of reported lithium metal anodes (Fig. R2).

Fig. R2. The comparison of the Coulombic efficiencies and area capacities of the Li||Cu cells in this work and literatures.

In addition, corrosion current density was also tested to demonstrate the relieved reactivity of Li metal towards the electrolyte (Supplementary Fig.13 and Fig. R3). The YP-Li demonstrates a low and stable corrosion current density (~ 0.3 mA cm⁻²) during whole cycling process, which suggests that YP-Li exhibits a low extent of side reactions during cycling. While the corrosion current density of bare Li experiences a substantial increase (from 0.62 to 1.42 mA cm⁻²), indicating that YP can relieve the reactivity between Li metal and electrolytes during cycling.

Supplementary Fig. 13. Potentiodynamic polarization curves with and without YP after 1, 20, and 50 cycles.

Fig. R3. The corrosion current density with YP-Li and bare Li electrodes after 1, 20, and 50 cycles.

Furthermore, as you mentioned, pressure is a very important parameter for lithium metal batteries, and it affects the cycle life. In our work, the pressure used for pouch cell was 250 kPa. This information has been added in the revised manuscript (page 28, line 2-3; page 31, line 5) and table of pouch cell parameters in the revised supplementary information (SI, page 19), which is also shown below.

“The long-term cycling stability is presented in Fig. 6e, in which the pouch cell is cycled for 150 cycles at 0.5 C (3 mA cm⁻²) under the pressure of 250 kPa with a high capacity retention of 87.42%.”

“After two formation cycles at 0.1 C (1 C=6 mA cm⁻²), the pouch cells were cycled under the same voltage range with 0.1 C charge and 0.5 C discharge under 25 °C at the pressure of 250 kPa.”

Reviewer Comment 4: *In Figure 1a, why the (110) peak shifts but (200) peak does not change during lithium plating? In Figure S8, why the (110) peak appears very sharp but the (200) peak is so broad? They are supposed to have similar peak width. It seems that Be window produces some peaks around the Li (200) peak. Would this undermine the basis of in situ XRD data analysis?*

Response:

Thanks a lot for your comments. In our study, *in situ* X-ray diffraction (XRD) was performed within a sealed chamber equipped with a Be window, which allows X-rays to penetrate through Be and reach the surface of the Li metal sample. In Supplementary Fig. 8, the (110) peak appears exceptionally sharp but the (200) peak exhibits broad. This phenomenon is attributed to the presence of two strong peaks of Be (Supplementary Fig. 9) near the (200) plane in the Li metal, making the peak of (200)

plane broad. However, given a defined sample, the contribution of the peak intensity from Be remains constant. Thus, the dominant peak intensity of the (200) plane upon lithium plating process in the YP-Li electrode, as shown in Supplementary Fig. 8d, undoubtedly arise from the Y-doped Li metal itself. This conclusion is also well supported by the analysis of *ex situ* XRD on lithium metal after different cycles, excluding the effect of Be window. As illustrated in Supplementary Fig. 10, the predominant XRD peak of bare Li corresponds to the (110) crystal plane after different cycles (Supplementary Fig. 10a). In contrast, the (200) plane becomes dominant for the YP-Li electrodes, indicating a preferred orientation shift from (110) to (200) upon cycling (Supplementary Fig. 10b). These results are entirely consistent with the *in situ* XRD findings.

Indeed, (110) peak shift a little bit (Fig. 1a), which was occurred in the process of *in situ* XRD test. We believe that this should be attributed to the stacking manner of lithium crystals grow during Li deposition process. Bare Li exhibits a dominant XRD peak belonging to the (110) crystal plane during Li plating process (Supplementary Fig. 8b), indicating that (110) is the preferential growth crystal plane.²² Li metal grows continuously on (110) plane upon prolonging Li plating time.²³ However, the chemical/electrochemical reduction between bare Li and electrolytes gives rise to an unstable SEI, leading to non-uniform stresses within the battery. This, in turn, may affect the growth and stacking of bare Li, ultimately resulting in a little bit shift of (110) peak. Additionally, surface irregularities of bare Li and environmental factors during *in-situ* testing also influence the peak shift.

In our research, we focus on variations in the relative intensity of the (110) and (200) peaks to further demonstrate changes in the preferred orientation. In Supplementary Fig. 8b, bare Li exhibits a dominant XRD peak belonging to the (110) crystal plane upon cycling. While the (200) plane becomes dominant for the YP-Li electrodes (Supplementary Fig. 8d), which indicates a change from (110) to (200) in the preferred orientation upon cycling.

Supplementary Fig. 8. (a, c) Typical discharge curves and (b, d) *In situ* XRD patterns of the bare Li and YP-Li.

Supplementary Fig. 9. XRD pattern of Be.

Supplementary Fig. 10. XRD patterns of the (a) bare Li and (b) YP-Li after different cycles.

Reference:

22. Shi, F., et al. Strong texturing of lithium metal in batteries. *Proc. Natl. Acad. Sci.* **114**, 12138-12143 (2017).
23. Lei, Z., et al. Corrosion performance of ZrN/ZrO₂ multilayer coatings deposited on 304 stainless steel using multi-arc ion plating. *Appl. Surf. Sci.* **431**, 170-176 (2018).

Reviewer Comment 5: *There is no description of how in situ XRD was carried out. This is an important experiment because authors mainly use its result to support the claim of (200) lithium deposition.*

Response:

According to your suggestion, we have supplemented the detailed procedure of *in situ* XRD in the *Methods Section* in the revised manuscript (page 32, line 17-22; page 33, line 1), which is also shown below.

“*In situ* X-ray diffraction (XRD) was recorded using Aeris within the range of 30-70°. The Li||Li symmetric cell was rested for 4 hours, and then it was cycled under a high areal capacity of 4 mAh cm⁻² with a current density of 1 mA cm⁻² on a LAND multichannel battery tester (CT2001A) with the simultaneous collection of XRD data upon Li plating. These cells, sealed in the cell molds containing beryllium (Be), were assembled in an argon-filled glovebox (H₂O<0.1 ppm, O₂<0.1 ppm).”

Reviewer Comment 6: *Authors claim that YP coating leads to 4 times less side reaction. How the magnitude of side reaction is measured? It cannot be simply indicated by the cycle life which is the overall outcome determined by side reactions, impedance, and many other factors.*

Response:

Thanks a lot for the reviewer’s comments. As you mentioned, side reactions cannot be simply indicated by the cycle life, which is the overall outcome determined by side reactions, impedance, and many other factors.^{24, 25} In our work, we quantified the magnitude of side reactions by measuring corrosion current densities after different cycles.²⁶ As shown in Supplementary Fig. 13 and Fig. R4, YP-Li demonstrates a low and stable corrosion current density (~0.30 mA cm⁻² even after 50 cycles) during whole cycling process, which suggests that YP-Li exhibits a low extent of side reactions during cycling. While the corrosion current density of bare Li experiences a substantial increase (from 0.62 mA cm⁻² after 1 cycle to 1.42 mA cm⁻² after 50 cycles). Therefore, YP can significantly reduce undesired side reactions between Li and electrolytes by 4 times (from 1.42 to ~0.30 mA cm⁻²), thus enabling a stable lithium-electrolyte interface. Furthermore, we also confirmed that the YP-Cu displays stable and small charge transfer resistance of ~ 16 Ω even after 50 cycles, while a dramatic resistance swing is found in bare Cu cells (from 18 to 42 Ω) (Supplementary Fig. 23). These results are

consistent with the corrosion current density, confirming the advantage of the YP in stabilizing the lithium/electrolyte interface.

Supplementary Fig. 13. Potentiodynamic polarization curves with YP-Li and bare Li electrodes after 1, 20, and 50 cycles.

Fig. R4. The corrosion current density with YP-Li and bare Li electrodes after 1, 20, and 50 cycles.

Supplementary Fig. 23. Nyquist plots for (a) YP-Cu and (b) bare Cu tested at different cycles.

References:

24. Zhang, Q.-K., et al. Homogeneous and mechanically stable solid–electrolyte interphase enabled by trioxane-modulated electrolytes for lithium metal batteries. *Nat. Energy* **8**, 725-735 (2023).
25. Huang, Z., et al. A salt-philic, solvent-phobic interfacial coating design for lithium metal electrodes. *Nat. Energy* **8**, 577-585 (2023).
26. Lei, Z., et al. Corrosion performance of ZrN/ZrO₂ multilayer coatings deposited on 304 stainless steel using multi-arc ion plating. *Appl. Surf. Sci.* **431**, 170-176 (2018).

Response to Reviewer #2

Reviewer #2:

In this manuscript, the authors report a synergetic regulation strategy of SEI mechanics and crystallographic orientation of Li via a YF₃/polymethyl methacrylate composite (YP) skin. Among them, a Y-doped Li metal can be generated by the reaction of YP with Li metal, and Y-doped (200) plane has the lowest surface energy, which enables the preferred oriented growth of Li metal along the (200) crystal plane. Thus, ultimately accomplishing dense and smooth deposition even at high-areal capacity of 10 mAh cm⁻². The authors also studied the SEI systematically. At last, a 4.2 Ah Li metal pouch cell reaches a high energy density of 468 Wh kg⁻¹ and undergoes 150 cycles. This work is interesting. However, the following comments have to be addressed before publication.

Response:

Thanks for your positive evaluation about our work. Based on your suggestions, the minor errors have been corrected in the revised manuscript. In addition, the responses to the questions are also listed point by point as shown below.

Reviewer Comment 1: *For the figure of the performance of high-energy-density (over 300 Wh kg⁻¹) Ah-type Li metal pouch cells, some recently published references are missing.*

Response:

Thanks for your comments. Following your suggestion, the recently published references [*Nat. Commun.* **2022**, *13*, 6788; *Nat. Commun.* **2023**, *14*, 4047; *Adv. Mater.* **2023**, *35*, 2211032; *Angew Chem. Int. Ed.* **2023**, *62*, e202306889.] have been supplemented in Fig. 6f and Supplementary Fig. 38 in the revised manuscript. The corresponding contents have been updated in the revised manuscript (page 26; page 27, line 2-4; page 28, line 7) and revised supplementary information (SI page 18), which is also shown below.

“More importantly, the high energy density (468 Wh kg⁻¹) and excellent stability (0.08% decay/cycle) of our designed YP-Li||NCM811 pouch cell outperform most of reported Li metal pouch cells (Fig. R5-6), especially under such extremely harsh testing

conditions.^{24, 27-41}»

Fig. R5. The performance of high-energy-density (over 300 Wh kg⁻¹) Ah-type Li metal pouch cells in published literatures and this work.

Fig. R6. The comparison performance of of high-energy-density (over 300 Wh kg⁻¹) Ah-type Li metal pouch cells in published literatures and this work.

References:

24. Zhang, Q.-K., et al. Homogeneous and mechanically stable solid–electrolyte interphase enabled by trioxane-modulated electrolytes for lithium metal batteries. *Nat. Energy* **8**, 725-735 (2023).
27. Niu, C., et al. High-energy lithium metal pouch cells with limited anode swelling and long stable cycles. *Nat. Energy* **4**, 551-559 (2019).
28. Qiao, Y., et al. A high-energy-density and long-life initial-anode-free lithium battery enabled by a Li₂O sacrificial agent. *Nat. Energy* **6**, 653-662 (2021).
29. Gao, Y., et al. Effect of the supergravity on the formation and cycle life of non-aqueous lithium metal batteries. *Nat. Commun.* **13**, 5 (2022).
30. Niu, C., et al. Balancing interfacial reactions to achieve long cycle life in high-energy lithium metal batteries. *Nat. Energy* **6**, 723-732 (2021).
31. Chang, Z., et al. An improved 9 micron thick separator for a 350 Wh/kg lithium metal rechargeable pouch cell. *Nat. Commun.* **13**, 6788 (2022).
32. Kwon, H., et al. Weakly coordinated Li ion in single-ion-conductor-based composite enabling low electrolyte content Li-metal batteries. *Nat. Commun.* **14**, 4047 (2023).
33. Huang, K., et al. Regulation of SEI formation by anion receptors to achieve ultra-stable

- lithium-metal batteries. *Angew. Chem. Int. Ed.* **60**, 19232-19240 (2021).
34. Zhang, M., et al. Boosting the temperature adaptability of lithium metal batteries via a moisture/acid-purified, ion-diffusion accelerated separator. *Adv. Energy Mater.* **12**, 2201390 (2022).
 35. Tan, Y.-H., et al. Lithium fluoride in electrolyte for stable and safe lithium-metal batteries. *Adv. Mater.* **33**, 2102134 (2021).
 36. Zhang, K., et al. A High-performance lithium metal battery with ion-selective nanofluidic transport in a conjugated microporous polymer protective layer. *Adv. Mater.* **33**, e2006323 (2021).
 37. Zhang, Y., et al. Enabling 420 Wh kg⁻¹ stable lithium metal pouch cells by Lanthanum doping. *Adv. Mater.* **35**, 2211032 (2023).
 38. Zhang, G., et al. A monofluoride ether-based electrolyte solution for fast-charging and low-temperature non-aqueous lithium metal batteries. *Nat. Commun.* **14**, 1081 (2023).
 39. Zhang, Q. K., et al. Reforming the uniformity of solid electrolyte interphase by nanoscale structure regulation for stable lithium metal batteries. *Angew. Chem. Int. Ed.* **62**, e202306889 (2023).
 40. Zhang, S., et al. In situ-polymerized lithium salt as a polymer electrolyte for high-safety lithium metal batteries. *Energy Environ. Sci.* **16**, 2591-2602 (2023).
 41. Mao, M., et al. Anion-enrichment interface enables high-voltage anode-free lithium metal batteries. *Nat. Commun.* **14**, 1082 (2023).

Reviewer Comment 2: *The characterization of Li metal in pouch cell should also be provided to tell the effect of the coating layers in practical cells.*

Response:

Thanks a lot for the reviewer's comments. Based on your suggestion, we supplemented the optical photographs, XRD pattern and SEM image of the cycled YP-Li anode in YP-Li||NCM811 pouch cell after 40 cycles. The surface of cycled YP-Li anode is compact (Fig. R7), indicating that YP composite skin could stabilize the Li-electrolyte interface and relieve the side reactions between Li metal and electrolytes, which were demonstrated by reduced and stable corrosion current density upon cycling (Supplementary Fig. 13). Furthermore, YP-Li shows a dominant XRD peak belonging to the (200) crystal plane (Fig. R8a), suggesting that (200) plane is the preferred orientation. The uniform and compact morphology is observed from the surface of the YP-Li anode (Fig. R8b). These results are consistent with it in coin cells. Therefore, YP composite skin tunes the preferential crystallographic orientation to (200) plane and induces the uniform Li deposition.

Fig. R7. Postmortem optical photographs of the YP-Li||NCM811 pouch cell after 40 cycles.

Fig. R8. (a) XRD pattern and (b) morphology of YP-Li anode after 40 cycles.

Reviewer Comment 3: Symmetric cells with different current density should also be provided to tell the performance.

Response:

We are grateful with the reviewer's comment. According to your suggestion, we have expanded the symmetrical Li||Li cells to investigate the Li plating/stripping performance of YP-Li anode, as shown in Supplementary Fig. 35-36. The corresponding contents have been updated in the revised manuscript (page 24, line 14-22; page 25, line 1-9) and revised supplementary information (SI page 16-17), which is also shown below.

“Symmetrical Li||Li cells were also employed to investigate the Li plating/stripping performance of YP-Li anode. As shown in Supplementary Fig. 35, the symmetrical cell with the YP-Li anode delivers Li plating/stripping overpotentials of 23, 37, 58, 76, and 89 mV with a fixed deposition capacity of 1 mAh cm⁻² when the current density increased from 0.25 to 0.5, 1, 2, and 3 mA cm⁻², respectively, which are much lower than that of cell with bare Li anode. Moreover, the YP-Li||YP-Li cell maintains long-term stability for 1200 h with 2 mAh cm⁻² at 1 mA cm⁻². By comparison, the cell with

bare Li anode displays significantly larger overpotential and experiences severe voltage oscillations (Supplementary Fig. 36a). When the current density increases to 2 mA cm^{-2} , YP-Li anode still demonstrates stable Li plating/stripping overpotentials for over 1000 h under 2 mAh cm^{-2} , whereas bare Li anode shows larger and more pronounced voltage fluctuations even in the initial cycles (Supplementary Fig. 36b). The striking differences in cycling performance strongly demonstrate the superiority of the YP protective layer in promoting dendrite-free Li plating/stripping behavior and maintaining a stable lithium-electrolyte interface.”

Supplementary Fig. 35. Voltage profiles of YP-Li||YP-Li and Li||Li symmetrical cells at various current densities with a fixed capacity of 1 mAh cm^{-2} .

Supplementary Fig. 36. Cycling stability of YP-Li||YP-Li and Li||Li symmetrical cells at different current density of (a) 1 mA cm^{-2} and (b) 2 mA cm^{-2} with an areal capacity of 2 mAh cm^{-2} .

Reviewer Comment 4: How could the authors ensure the uniform distribution of the YP in PMMA? Is it small particles, clusters or molecules?

Response:

Thanks a lot for the reviewer's comments. Following your suggestion, we have supplemented morphology and particle size distribution of YF_3 particles, and the top-view SEM image of YP-Cu electrode, as shown in Fig. R8. The mean size of YF_3 particles is $\sim 2 \mu\text{m}$ (Fig. R9a-b). The homogeneous YP solution was evenly coated on Cu foil through blade casting by adjustable coater on an automatic coating machine. The top-view SEM images of the YP-Cu electrode (Fig. R9c) and uniform distribution of C, O, Y, and F elements (Supplementary Fig. 2) further proves that YF_3 particles are uniformly distributed in PMMA, and YP are evenly covered on the Cu foil. Additionally, the corresponding contents have been updated in the revised manuscript (page 8, line 15-17).

Fig. R9. (a) Morphology and (b) particle size distribution of the YF_3 particles. (c) Top-view SEM images of the YP-Cu electrode.

Supplementary Fig. 2. The EDS element mappings of C, O, Y, and F in the YP-Cu electrode.

Reviewer Comment 5: “Therefore, it can be concluded that Y-doped Li metal can reduce the reactivity of Li metal with electrolytes, thus enabling a stable lithium-

electrolyte interface.” How can the author make sure that this effect comes from the Y doping rather than the protection of PMMA?

Response:

We are thankful to the comment. We supplemented the corrosion current density of PMMA-Li electrodes after different in the revised manuscript (page 11) and revised supplementary information (SI page 7). Subsequently, we compared the corrosion current density of PMMA-Li electrodes (Supplementary Fig. 14) with that of YP-Li and bare Li (Supplementary Fig. 13) to demonstrate the effect of Y doping and PMMA protective layer on reducing the reactivity of Li metal with electrolytes. The corresponding contents have been updated in the revised manuscript (page 13, line 20-22; page 14, line 1-8), which is also shown below.

“To further investigate whether the reducing undesired side reactions is attributed to the Y doping or PMMA, we also designed a control sample with only PMMA on the Li surface, where the corrosion current density of PMMA-Li is 0.55, 0.64, and 0.89 mA cm⁻² after 1, 20, and 50 cycles, respectively (Supplementary Fig. 14). These values are lower than that of bare Li but higher than YP-Li (Fig. 1e), which is attributed to the protection of PMMA. This protection contributes to shielding the lithium metal from undesirable side reactions with electrolytes.¹⁸ Therefore, it can be concluded that the synergistic effect of Y-doped Li metal and PMMA protective layer reduces the reactivity of Li metal with electrolytes, thus enabling a stable lithium-electrolyte interface.”

Supplementary Fig. 14. Potentiodynamic polarization curves with PMMA-Li electrodes after 1, 20, and 50 cycles.

Supplementary Fig. 13. Potentiodynamic polarization curves with YP-Li and bare Li electrodes after 1, 20, and 50 cycles.

Fig. 1e. The corrosion current density of YP-Li, PMMA-Li, bare Li electrodes after 1, 20, and 50 cycles.

Reference:

- Zhou, Z., et al. A robust, highly stretchable ion-conductive skin for stable lithium metal batteries. *Chem. Eng. J.* **396**, 125254 (2020).

Reviewer Comment 6: “These changes effectively passivate Li metal, thereby significantly reducing undesired side reactions between Li and electrolytes by 4 times.” Again, how can the author make sure that this effect comes from the Y doping rather than the protection of PMMA?

Response:

Thanks for the reviewer’s comment. In this work, we demonstrate the *in-situ* generation of Y-doped lithium metal through the reaction of YP with Li metal, which reduces the surface energy of the (200) plane, and tunes the preferential crystallographic orientation to (200) plane from conventional (110) plane during Li plating. Benefiting from the lowest-surface-energy (200) planes exposed to the electrolyte, the reactivity of Li metal towards the electrolyte is relieved, which can promote the formation of a stable lithium-electrolyte interface, as demonstrated by reduced corrosion current

density.²⁶ As shown in Supplementary Fig. 13, the YP exhibits a lower corrosion current density ($\sim 0.30 \text{ mA cm}^{-2}$) than bare (1.42 mA cm^{-2}) even after 50 cycles, suggesting that YP can significantly reduce undesired side reactions between Li and electrolytes by 4 times.

Furthermore, based on your suggestions, to investigate whether the reducing undesired side reactions is attributed to the Y doping or PMMA, we supplemented the corrosion current density of PMMA-Li electrodes after different cycles in the revised manuscript (Fig. 1e) (page 11) and revised supplementary information (Supplementary Fig. 14) (SI page 7), and compared the corrosion density of PMMA-Li electrodes with that of YP-Li and bare Li. The corresponding contents have been updated in the revised manuscript (page 13, line 20-22; page 14, line 1-8), which is also shown below.

“To further investigate whether the reducing undesired side reactions is attributed to the Y doping or PMMA, we also designed a control sample with only PMMA on the Li surface, where the corrosion current density of PMMA-Li is 0.55, 0.64, and 0.89 mA cm^{-2} after 1, 20, and 50 cycles, respectively (Supplementary Fig. 14). These values are lower than that of bare Li but higher than YP-Li (Fig. 1e), which is attributed to the protection of PMMA. This protection contributes to shielding the lithium metal from undesirable side reactions with electrolytes.¹⁸ Therefore, it can be concluded that the synergistic effect of Y-doped Li metal and PMMA protective layer reduces the reactivity of Li metal with electrolytes, thus enabling a stable lithium-electrolyte interface.”

Supplementary Fig. 13. Potentiodynamic polarization curves with YP-Li and bare Li electrodes after 1, 20, and 50 cycles.

Supplementary Fig. 14. Potentiodynamic polarization curves with PMMA-Li electrodes after 1, 20, and 50 cycles.

Fig. 1e. The corrosion current density of YP-Li, PMMA-Li, bare Li electrodes after 1, 20, and 50 cycles.

References:

18. Zhou, Z., et al. A robust, highly stretchable ion-conductive skin for stable lithium metal batteries. *Chem. Eng. J.* **396**, 125254 (2020).
26. Lei, Z., et al. Corrosion performance of ZrN/ZrO₂ multilayer coatings deposited on 304 stainless steel using multi-arc ion plating. *Appl. Surf. Sci.* **431**, 170-176 (2018).

Reviewer Comment 7: *There are some minor errors in the manuscript, such as page 12 figure 2e, SI page 9 figure S8...*

Response:

Sorry for our careless mistakes. We have corrected the minor errors in the revised manuscript (page 13, line 13-14) and supplementary information (SI page 5).

“Supplementary Fig. 13-14 and Fig. 1e show that YP-Li demonstrates a low and stable corrosion current density ($\sim 0.30 \text{ mA cm}^{-2}$) during whole cycling process (Supplementary Fig. 13), which suggests that the YP-Li exhibits a low extent of side reactions during cycling.”

“Supplementary Fig. 8. (a, c) Typical discharge curves and (b, d) *In situ* XRD patterns of the bare Li and YP-Li. (e-f) *In situ* synchrotron XRD contour map during Li

deposition of the bare Li and YP-Li.”

Response to Reviewer #3

Reviewer #3:

These authors report on a Yttrium fluoride-polymer composite that appears to form favorable SEI on the Li metal anode. Authors found that the YF_3 reacts with Li metal to form LiF and a Yttrium-doped Li material that alter the surface energetics and crystallographic growth features for deposited Li. On these basics authors claim to have created an energy dense Li cell that according to the paper title delivers 460 Wh/kg. I have a number of concerns that make it less clear about the importance of the study for Nature Communications.

Response:

We deeply appreciate the reviewer’s careful reading and helpful comments for our manuscript. The responses to the questions are listed point by point as shown below.

***Reviewer Comment 1:** The concept of using a particular/special additive in electrolyte or at interface to improve cycling of Li is not new or novel. There are many papers of this type in the literature and few have so far actually led to advances. What would be of interest to the readership is the underlying principle by which additive works and how it can be extended to other additives.*

Response:

We are grateful to the reviewer’s comments. As you mentioned, many research studies have been devoted to improving cycling performance of lithium metal batteries by modifying the interface and introducing functional electrolyte additives.^{19, 24, 25, 39, 42} Although the considerable success achieved to a certain extent at low Li plating capacities (<2 mAh cm^{-2}), challenges still exist at high deposition capacities (≥ 4 mAh cm^{-2}), which is essential for achieving high-energy-density and long-cycling-life lithium metal batteries. More importantly, current strategies primarily focus on modifying the chemical composition of the interface to inhibit Li dendrite growth. However, this approach does not fundamentally address the issues of dendrite growth

and unstable interfaces.

In our work, we emphasize the critical yet less-studied perspective, synergetic regulation of SEI mechanics and crystallographic orientation of Li metal anode via a YF₃/PMMA (YP) composite skin to fundamentally stabilize the interface and suppress dendrite growth **under high deposition capacity**. We demonstrate that Y doping enables a preferred crystal plane change to (200) from (110) plane in the plating process. This crystallographic orientation changes effectively relieve the reactivity between Li metal and electrolytes during cycling, thus achieving a dense deposition **even at high-areal-capacity of 10 mAh cm⁻²**. Moreover, YP layer with suitable modulus (~1.02 GPa) can enhance mechanical stability and maintain structural stability of SEI upon cycling. More attractively, a 4.2 Ah Li metal pouch cell, **under a low N/P ratio of 1.67, lean electrolyte of 1.98 Ah g⁻¹, and high current density of 3 mA cm⁻²**, reaches a high energy density of 468 Wh kg⁻¹ and undergoes 150 cycles. The significances of this work are listed as below:

(1) Different from previous reported studies, we report a synergetic regulation of SEI mechanics and crystallographic orientation of Li metal anode, which renders mechanical stability of SEI and dense deposition of Li metal even at high-areal-capacity of 10 mAh cm⁻². These results are attributed to the change in the preferred orientation of Li metal and the reduction in reactivity between Li metal and electrolytes. More importantly, YP layer with a suitable modulus of ~1.02 GPa serves as a protective sheath to enhance mechanical stability and uniform stress distribution of SEI, thus preventing its structure from being destroyed upon cycling.

(2) We demonstrate a 4.2 Ah pouch cell with a high energy density of 468 Wh kg⁻¹ and exceptional cycling stability (0.08% decay/cycle) based on the YP-Li anode under lean electrolyte (1.98 g Ah⁻¹) and low N/P ratio of 1.67, which is superior to the reported lithium metal batteries (see Table R1 and Fig. R10 below). We believe our findings will greatly contribute to the practical applications of high-energy-density lithium metal batteries.

In the revised manuscript, we give more descriptions to better elaborate the highlights (page 7, line 1-11; page 8, line 1-14) and hopefully our proposed novel strategy can

boost the practical application of lithium metal batteries.

Table R1. The comparison of Li metal pouch cells in this work and literatures.

Energy Density (Wh kg ⁻¹)	Decay rate per cycle	N/P ratio	E/C ratio (g Ah ⁻¹)	Capacity (Ah)	Refs.
468	0.08%	1.67	1.98	4.2	This Work
440	0.063%	1.8	2.1	5.3	Ref.1 Nat. Energy 2023, 8, 725-735
426	0.1%	2	2.4	0.32	Ref.2 Nat. Commun. 2023, 14, 1081
442	0.2%	0	2.75	0.22	Ref.3 Nat. Commun. 2023, 14, 1082
437	0.077%	2.7	2	2.8	Ref.4 Energy Environ. Sci. 2023, 16, 2591-2602
325	/	2.6	2.7	2.5	Ref.5 Nat Commun. 2022, 13, 5
350	0.04%	1	2.4	2	Ref.6 Nat. Energy 2021, 6, 723-732
323	0.067%	1.75	2.5	2.46	Ref.7 Nat. Energy 2021, 6, 653-662
300	0.07%	2.6	3.0	1	Ref.8 Nat. Energy 2019, 4, 551-559
357	0.096%	2.54	3	2.5	Ref.9 Angew. Chem. Int. Ed. 2021, 60, 19232-19240
380	0.081%	2.5	2.9	3.45	Ref.10 Adv. Mater. 2021, 33, 2102134
400	0.285%	1.34	2.5	10	Ref.11 Adv. Mater. 2021, 33, 2006323
414	0.093%	2.2	2.3	3.22	Ref.12 Adv. Energy Mater. 2022, 12, 2201390.

Fig. R10. The comparison performance of of high-energy-density (over 300 Wh kg⁻¹) Ah-type Li metal pouch cells in published literatures and this work.

References:

19. Xie, Y., et al. Surface modification using heptafluorobutyric acid to produce highly stable Li metal anodes. *Nat. Commun.* **14**, 2883 (2023).
24. Zhang, Q.-K., et al. Homogeneous and mechanically stable solid–electrolyte interphase enabled by trioxane-modulated electrolytes for lithium metal batteries. *Nat. Energy* **8**, 725-735 (2023).
25. Huang, Z., et al. A salt-philic, solvent-phobic interfacial coating design for lithium metal electrodes. *Nat. Energy* **8**, 577-585 (2023).
39. Zhang, Q. K., et al. Reforming the uniformity of solid electrolyte interphase by nanoscale structure regulation for stable lithium metal batteries. *Angew. Chem. Int. Ed.* e202306889 (2023).
42. Zhao, Y., et al. Electrolyte engineering via ether solvent fluorination for developing stable non-aqueous lithium metal batteries. *Nat. Commun.* **14**, 299 (2023).

Reviewer Comment 2: Why Yttrium is used in first place? YF₃ is expensive (more than

2000 USD/kg), which would make battery that depend on it impractical. If authors are correct that the reason for the high capacity and long life is that the Y-doped Li favors the (200) facet, same approach might be adopted to another less so expensive chemistry. It is a serious weakness that authors make no effort to show this.

Response:

Thanks for the reviewer's comment. To achieve a highly stable Li metal anode under high deposition capacity conditions, addressing the challenges of uncontrollable dendrite growth and unstable Li-electrolyte interface is essential. Fundamentally, the formation of Li dendrites undergoes two steps involving nucleation and growth of the Li crystal.¹ As with other metal anodes, Li metal grows continuously on its early-formed nuclei during cycling. The final deposition morphology tightly relies on the preferential crystallographic orientation of Li metal crystal.^{2,3}

The preferred crystallographic orientation depends on the surface energy of the crystal.^{7, 8} Doping with rare-earth element is considered as an efficient strategy to change the surface energy and thus control the preferred crystallographic orientation.^{9, 10} Theoretical studies have demonstrated that Li metal with low surface energy result in high interfacial energy, thereby promoting the horizontal migration of Li⁺ and effectively suppressing the vertical growth of Li metal. In addition, the lower surface energy also leads to larger Li deposit particles, which is beneficial for alleviating the reactivity between Li metal and electrolytes.^{11, 12} Considering yttrium's favorable affinity with Li, along with its large atomic radius,¹³ we hypothesized that doping Y into Li metal could tune and reduce the surface energy of Li metal, and thus regulate the preferred orientations, ultimately increasing the nucleation radius of Li metal. This fundamentally further reduces undesired side reactions between Li and electrolytes and inhibits Li dendrite growth. Based on the aforementioned hypothesis, we selected Y as a representative sample to conduct a series of experiments and characterizations to validate our conjecture.

Actually, the YF₃ used in our study is inexpensive; it was purchased from Macklin Biochemical Co., Ltd (Shanghai, China), with a mere cost ~200 USD for 500 g ([http://www.macklin.cn/search/YF₃](http://www.macklin.cn/search/YF3)). In our work, the YP mixture was obtained by

mixing YF₃ particles (5 mg) with PMMA (5 mg) with a weight ratio of 1:1 in THF (10 g) and stirring overnight. The application amount was 2 μL cm⁻². Therefore, in the pouch cell, the dosage of YP composite skin was ~1 mL with a cost of ~0.0002 USD. More attractively, we demonstrate a 4.2 Ah YP-Li||NCM811 pouch cell with a high energy density of 468 Wh kg⁻¹ and remarkable capacity stability of 0.08% decay/cycle based on the YP-Li anode under realistic conditions, making it more promising for practical application. In addition, we will further explore the effects of other more economical compounds with similar properties in lithium metal batteries in the subsequent work.

***Reviewer Comment 3:** There are already previous studies that show that SEI composed of F-rich species enhance cycle life of the Li anode. Authors claims that a different mechanism is responsible for results reported but this conclusion is not strong because for every mole of YF₃ 3 moles of LiF are made. The special role claimed for the Y-doped Li would be more convincing if authors showed that Yttrium salts that do not contain F achieve the similar results.*

Response:

We really appreciate the reviewer's comment. In our study, we report a synergistic regulation strategy of SEI mechanics and crystallographic orientation of Li via a YF₃/PMMA (YP) composite skin to fundamentally realize mechanical stability of SEI and dense deposition of Li metal. Consequently, a 4.2 Ah Li metal pouch cell reaches a high energy density of 468 Wh kg⁻¹ and outstanding cycling stability of 0.08% decay/cycle under harsh condition. These competitive results are attributed to the synergistic effect between SEI mechanics and crystallographic orientation of Li.

Furthermore, based on your suggestion, we assembled and evaluated asymmetric Li||Cu cells and coin-type Li||NCM811 full cells using non-fluorinated yttrium salts of YCl₃ (YP1) under the same conditions as the YP. As a result, the Li||YP1-Cu cells exhibit long-term cycling stability up to 450 cycles and high average CEs of 99.19% with 2 mAh cm⁻² at 2 mA cm⁻² (Fig. R11a). The corresponding voltage profiles are displayed in Fig. R11b at various cycles, where YP1-Cu exhibits flat and smooth

voltage plateaus with a low initial overpotential of ~ 103 mV followed by slightly decrease to 89 mV in the 50th cycle, and maintains low values (88 mV) with almost no change in the subsequent cycles, indicating the constant formation of stable interface. For comparison, Li||YP1-Cu cells exhibit similar electrochemical stability with Li||YP-Cu cells.

Fig. R11. (a) Coulombic efficiencies of YP1-Cu electrodes with areal capacities of 2 mAh cm^{-2} at a current density of 2 mA cm^{-2} . (b) The voltage profiles of YP1-Cu electrodes with areal capacities of 2 mAh cm^{-2} at a current density of 2 mA cm^{-2} .

Fig. R12. (a) Long-term cycling performance of the coin-type full cells with YP1-Li anode at 1 C. (b) The corresponding charge-discharge voltage profiles of YP1-Li||NCM811 full cells.

Fig. R12a shows the cycling performance of full cells with a high-loading NCM811 ($\sim 20 \text{ mg cm}^{-2}$, $\sim 4 \text{ mAh cm}^{-2}$) under lean electrolyte of $10 \mu\text{L mAh}^{-1}$ and thin lithium with the thickness of $20 \mu\text{m}$. YP1-Li||NCM811 cells can maintain excellent cycling stability with capacity retention up to 96.5% after 100 cycles at 1 C (4 mA cm^{-2}).

Additionally, the YP1-Li||NCM811 cell displays stable voltage plateaus with low voltage polarization during cycling (Fig. R12b). Anyway, Yttrium salts that do not contain F, such as YCl_3 , achieve the similar results to those obtained with YF_3 .

To further demonstrate the effect of Y doping into Li metal on mechanistic study, the tests related to the mechanism in our paper were also supplemented using YP1, including characteristics of the preferred orientation growth, corrosion current, and lithium deposition behavior.

Firstly, X-ray diffraction (XRD) was performed to track the crystal structure evolution of cycled Li metal anodes with YP1-Li electrodes in the same conditions as YP-Li electrodes. In our study, bare Li shows a dominant XRD peak belonging to the (110) crystal plane after different cycles (Supplementary Fig. 10a), suggesting that (110) is the main exposed crystal plane. In comparison, the (200) plane becomes dominant for the YP1-Li electrodes, which indicates a change from (110) to (200) in the preferred orientation (Fig. R13). Therefore, YP1-Li induces a shift in the preferred orientation of Li metal from (110) to (200) crystal plane during the Li deposition process, aligning with the phenomenon observed in YP-Li (Supplementary Fig. 10b). Corrosion current density was also tested to demonstrate the relieved reactivity of Li metal towards the electrolyte (Fig. R14). The YP1-Li shows a low and stable corrosion current density ($\sim 0.32 \text{ mA cm}^{-2}$) during whole cycling process, which suggests that the YP1 exhibits a low extent of side reactions during cycling. While the corrosion current density of bare Li experiences a substantial increase (from 0.62 to 1.42 mA cm^{-2}). This significant difference reveals that YP1 can reduce the reactivity of Li metal with electrolytes, thus enabling a stable lithium-electrolyte interface. These results are consistent with YP-Li electrodes.

Supplementary Fig. 10. XRD patterns of the (a) bare Li and (b) YP-Li after different cycles under high areal capacity of 4 mAh cm^{-2} with a current density of 1 mA cm^{-2} .

Fig. R13. Characteristics of the preferred orientation growth. XRD patterns of the YP1-Li after different cycles under high areal capacity of 4 mAh cm^{-2} with a current density of 1 mA cm^{-2} .

Fig. R14. Potentiodynamic polarization curves with the YP1-Li electrode after 1, 20, and 50 cycles and corresponding corrosion current density with YP-Li and bare Li electrodes after 1, 20, and 50 cycles.

Moreover, we also study lithium deposition behavior with YP1-Cu electrodes. As shown in Fig. R15, the SEM image of Li plating at different areal capacities (from 1 to

10 mAh cm⁻²) is provided. At the deposition capacity of 1 mAh cm⁻², YP1-Cu shows a rounded-like morphology (Fig. R15a), which gradually grows into a flat and smooth sheet-like structure as the deposition capacity increases (Fig. R15b). As it increased to 10 mAh cm⁻² (Fig. R15c), uniform and dense morphologies are observed.

Synchronously, the cross-sectional morphologies of the Li deposited on YP1-Cu electrodes were also scrutinized, as shown in Fig. R15d-f. The theoretical thickness of plating capacity for 1, 4, and 10 mAh cm⁻² is 5, 20, and 50 μm, respectively.⁴³ On YP1-Cu, the thickness values are 5.2, 22.5, and 52.8 μm at deposition capacities of 1, 4, and 10 mAh cm⁻², close to the theoretical value. The above series of experiments demonstrate that YP1-Cu can effectively induce the uniform lithium deposition and inhibit the dendrites growth. After 20 cycles, the deposited Li shows a smooth and compact morphology for the YP1-Cu electrode (Fig. R16). The dense structure is maintained even after 40 cycles (Fig. R16), further confirming the critical role of YP1 in stabilizing the interface during Li plating/stripping process. The above deposition behavior is consistent with it in YP-Cu electrodes.

Fig. R15. Morphologies of the Li||Cu half cells after first cycle with the YP1-Cu electrode. (a-c) Top-view SEM images of plating Li on YP1-Cu at a deposition capacity of 1, 4, and 10 mAh cm⁻² under a current density of 1 mA cm⁻² after the first cycle. (d-f) Cross-view SEM images of plating Li on YP1-Cu at a deposition capacity of 1, 4, and 10 mAh cm⁻² under a current density of 1 mA cm⁻² after the first cycle.

Fig. R16. The morphologies of Li deposited on the YP1-Cu electrode after 20 and 40 cycles. (a-b) Top view morphology of Li deposition on the YP1-Cu electrode at 1 mA cm^{-2} for 4 mAh cm^{-2} after 20 cycles. (c-d) Cross-section morphology of Li deposition on the YP1-Cu electrode at 1 mA cm^{-2} for 4 mAh cm^{-2} after 40 cycles.

References:

43. Fang, C., et al. Pressure-tailored lithium deposition and dissolution in lithium metal batteries. *Nat. Energy* **6**, 987-994 (2021).

Reviewer Comment 4: *The DFT analysis assume that the electrolyte does play a role. This is likely untrue for a reactive metal like Li where SEI formation will lower the effectiveness of a templating coating as the Li deposit thickness increases. Said differently, if the Y-doping can indeed control the crystal mode for Li in the first layer, is it really possible to completely passivate the subsequent Li deposit from reaction with electrolyte? Also, is it really possible to achieve dominantly 200 growth at 100 μm Li thicknesses as claimed. To support authors claims we would need to see current vs time data for the corrosion reaction measured at different Li thickness.*

Response:

We appreciate the reviewer's comments. In this work, we report a synergetic regulation strategy of SEI mechanics and crystallographic orientation of Li via a YF_3/PMMA (YP) composite skin to fundamentally realize mechanical stability of SEI and dense deposition of Li metal. Y doping enables a preferred crystal plane change to (200) from (110) plane in the plating process, which effectively decreases the reactivity between Li metal and electrolytes during cycling, thus achieving a dense deposition. Moreover, YP layer with suitable modulus ($\sim 1.02 \text{ GPa}$) can enhance mechanical

stability and maintain structural stability of SEI upon cycling. Therefore, even at high areal capacity of 4 mAh cm⁻², our synergetic regulation strategy can simultaneously achieve dense deposition in the bottom layer and mechanical stability in the top layer.

Furthermore, it is well known that there are many factors affecting the decomposition of the electrolyte on the Li metal surface, such as the lithium salt, solvent, the composition and structure of SEI, and the properties of Li metal itself, etc.^{14, 44, 45} Surface energy is an important factor affecting electrolyte decomposition. If the lowest-surface-energy planes exposed to the electrolyte, the reactivity of Li metal towards the electrolyte will be relieved.²³ In our work, when Y doped on the surface or the second layer of Li metal, Y-doped (200) crystal plane has the lowest surface energy, even lower than the (110) crystal plane (Supplementary Table 1). It reveals that Y doping intrinsically reduces the surface energy, enabling the (200) crystal plane becomes the preferred orientation and forming a stable interface.⁴⁶ Moreover, we also confirmed that the YP-Li demonstrates a low and stable corrosion current density (~0.30 mA cm⁻²) than bare Li (from 0.62 to 1.42 mA cm⁻²) during whole cycling process as demonstrated by corrosion current density (Supplementary Fig. 13 and Fig. R17). This significantly reveals that Y-doped Li metal can reduce the reactivity of Li metal with electrolytes, thus enabling a stable lithium-electrolyte interface. These results are consistent with the surface energy.

Supplementary Table 1. Total energy for the specific surface in the slab structures.

	Atom	E_{total} (eV/atom)	S (Å²)	\bar{E}_S (meV/Å²)	
	Li(110)	96	-1.8171	136.1426	31.30
	Li(211)	96	-1.7424	227.7913	34.17
	Li(200)	96	-1.7856	195.3471	39.17
	Li₉₅Y(110)1st	96	-1.8511	136.1426	33.48
	Li₉₅Y(211)1st	96	-1.7740	227.7913	36.57
	Li₉₅Y(200)1st	96	-1.8205	195.3471	20.35
	Li₉₅Y(110)2nd	96	-1.8574	136.1426	28.99
	Li₉₅Y(211)2nd	96	-1.7792	227.7913	34.40
	Li₉₅Y(200)2nd	96	-1.8230	195.3471	19.14

Supplementary Fig. 13. Potentiodynamic polarization curves with YP-Li and bare Li electrodes after 1, 20, and 50 cycles.

Fig. R17. The corrosion current density with YP-Li and bare Li electrodes after 1, 20, and 50 cycles.

According to your suggestions, we have supplemented the data on the corrosion current density versus cycle numbers at different Li thickness of 20, 50, and 100 μm . As depicted in Fig. R18-19, with an increase in cycle numbers and Li thickness, YP-Li consistently demonstrates a low and stable corrosion current density, whereas the corrosion current density of bare Li experiences a pronounced increase. These results indicate that YP can also mitigate the reactivity of Li metal with electrolytes even under various Li thickness during the cycling process.

Fig. R18. Potentiodynamic polarization curves with YP-Li and bare Li electrodes at different Li thickness of 20, 50, and 100 μm after 1, 20, and 50 cycles.

Fig. R19. The corrosion current density with YP-Li and bare Li electrodes at different Li thickness of 20, 50, and 100 μm after 1, 20, and 50 cycles.

References:

14. Liu, Y., et al. Self-assembled monolayers direct a LiF-rich interphase toward long-life lithium metal batteries. *Science* **375**, 739-745 (2022).
23. Lei, Z., et al. Corrosion performance of ZrN/ZrO₂ multilayer coatings deposited on 304 stainless steel using multi-arc ion plating. *Appl. Surf. Sci.* **431**, 170-176 (2018).
44. Yu, Z., et al. Molecular design for electrolyte solvents enabling energy-dense and long-cycling lithium metal batteries. *Nat. Energy* **5**, 526-533 (2020).
45. Wang, H., et al. Liquid electrolyte: The nexus of practical lithium metal batteries. *Joule* **6**, 588-616 (2022).

46. Wang, S., et al. Preferentially oriented Ag-TiO₂ nanotube array film: An efficient visible-light-driven photocatalyst. *J. Hazard. Mater.* **399**, 123016 (2020).

Reviewer Comment 5: *Is specific energy value stated in the title and reported in Figure 6 theoretical values? I see that the 468 Wh/kg value in Figure 6 is actually based on the first cycle capacity during formation (C/10) cycling. The value is lower by the 4th cycle and I estimate a cell specific energy close to that in Ref 51. At the very least, authors will need to fully explain how specific energy values given in Figure 6 are determined and under what conditions the new YP-Li result are superior to ref 51.*

Choice of PMMA polymer is not clear? Could other polymer work just as well? If not, explain this.

Response:

Thanks a lot for the reviewer's comments. It is well known that the cell-level specific energy is calculated by multiplying the total capacity and the cell voltage (taken as the mid-point voltage in discharge), and then dividing by the total mass of the pouch cell, including cathode, anode, electrolyte, separator, Al foil, package and lugs. In this work, an initial discharge capacity of 4.2 Ah was achieved when cycling at 0.1 C. The energy density is calculated as below (Table R2), where the total mass includes all components (cathode, anode, electrolyte, separator, Al foil, package and lugs). The corresponding calculation formula was also updated in the revised supplementary information (SI page 19, Supplementary Table 2).

$$\begin{aligned} \text{Energy density} &= \frac{\text{Capacity(Ah)} \times \text{Voltage (V)}}{\text{Mass (kg)}} \\ &= \frac{4.2 \text{ Ah} \times 3.8 \text{ V}}{0.03407 \text{ (kg)}} \\ &= 468 \text{ Wh kg}^{-1} \end{aligned}$$

Table R2. The detailed parameters in the pouch cell.

Capacity (Ah)- 4th cycle	Mid-value voltage (V)	Current density (mA cm ⁻²)	Total Weigh (g)	Energy density (Wh/kg)
4.0	3.8	0.6	34.07	468

We also calculate the energy density in the 4th cycle shown in **Table R3:**

$$\begin{aligned} \text{Energy density} &= \frac{\text{Capacity(Ah)} \times \text{Voltage (V)}}{\text{Mass (kg)}} \\ &= \frac{4.0 \text{ Ah} \times 3.8 \text{ V}}{0.03407 \text{ (kg)}} \\ &= 446 \text{ Wh kg}^{-1} \end{aligned}$$

Table R3. The detailed parameters in the pouch cell.

Capacity (Ah)- 4th cycle	Mid-value voltage (V)	Current density (mA cm⁻²)	Total Weigh (g)	Energy density (Wh/kg)
4.0	3.8	3	34.07	446

For Ref.51 [*Nat. Commun.* 2023, 14, 1082], anode-free pouch cells were constructed with high cathode loading of 4.64 mAh cm⁻² and lean electrolyte absorbance of 2.75 g Ah⁻¹. The two-layer pouch cells with a high capacity of 220 mAh, corresponding to the energy density of 365.9 Wh kg⁻¹, can achieve a capacity retention of 80% after 100 cycles at 0.1 C charge and 0.5 C discharge rate. Adapting the parameters of practical 10-layer pouch cells, the cell-level-specific energy of anode-free pouch cells is calculated to be 442.5 Wh kg⁻¹. The pouch cell in our work comprises a high-areal-capacity NCM811 cathode (6 mAh cm⁻² on each side) and an ultrathin YP-Li foil (100 μm thick, 10 mAh cm⁻² on each side), giving a N/P ratio of only 1.67. Meanwhile, the E/C ratio was limited at 1.98 g Ah⁻¹. Impressively, the assembled YP-Li||NCM811 pouch cell with a high initial capacity of 4.2 Ah delivers a high cell-level energy of 468 Wh kg⁻¹ when cycling at **0.1 C (~0.6 mA cm⁻²)**, which is superior to Ref. 51. The YP-Li||NCM811 pouch cell presents a discharge capacity of 4.0 Ah in the 4th cycle at **0.5 C (~3 mA cm⁻²)**, corresponding to the energy density of 446 Wh kg⁻¹, which is close to that in Ref. 51. Furthermore, YP-Li||NCM811 pouch cell delivers excellent cycling stability with a with a high capacity retention of 87.42% after 150 cycles even under tightly restricted conditions, including **N/P ratio is 1.67, E/C ratio is 1.98 g Ah⁻¹**. If the E/C ration is further reduced to 1.4 Ah g⁻¹, the energy density can reach 505 Wh kg⁻¹. Noticeably, the high energy density (**468 Wh kg⁻¹**) and excellent stability (**0.08% decay/cycle**) of our designed YP-Li||NCM811 pouch cell outperform most of reported Li metal pouch cells (Table R4 and Fig. R20), especially under such extremely harsh

testing conditions.

Table R4. The comparison of Li metal pouch cells in this work and literatures.

Energy Density (Wh kg ⁻¹)	Charge current (mA cm ⁻²)	Decay rate per cycle	N/P ratio	E/C ratio (g Ah ⁻¹)	Capacity (Ah)	Refs.
468	0.6	0.08%	1.67	1.98	4.2	This Work
442	0.46	0.2%	0	2.75	0.22	Nat. Commun. 2023, 14, 1082
440	0.28	0.063%	1.8	2.1	5.3	Nat. Energy 2023, 8, 725-735
437	0.74	0.077%	2.7	2	2.8	Energy Environ. Sci. 2023, 16, 2591-2602
430	0.28	0.029%	1.8	2.0	5.3	Angew. Chem. Int. Ed. 2023, e202306889
426	0.8	0.1%	2	2.4	0.32	Nat. Commun. 2023, 14, 1081
425	0.58	0.098%	1.73	1.76	3.1	Adv. Mater. 2023, 35, 2211032
418	0.57	0.0114%	1.75	2.5	5.13	Adv. Energy Mater. 2022, 12, 2200568
414	0.46	0.093%	2.2	3.22	3.2	Adv. Energy Mater. 2022, 12, 2201390

Fig. R20. The comparison performance of high-energy-density (over 400 Wh kg⁻¹) Ah-type Li metal pouch cells in published literatures and this work.

The mechanical stability of the SEI is much essential for achieving high-energy-density and long-cycling-life lithium metal batteries.⁴⁷ The organic components within the SEI are generally flexible to withstand moderate volumetric deformation upon cycling. Furthermore, the introduction of chains that interact with Li⁺ in the polymer layer facilitates Li⁺ transport at the interface. Consequently, applying a polymer layer on the Li anode contributes to enhance mechanical stability of SEI and promote the long-term operation of lithium metal batteries.⁴⁸

PMMA with excellent polymeric segmental motion, owns a high ionic conductivity and elastic modulus. The carbonyl of PMMA exhibits strong interactions with Li⁺, which could regulate uniform Li deposition.^{17, 18} Therefore, PMMA can serve as a protective sheath to enhance mechanical stability and uniform stress distribution of SEI, thus preventing its structure from being destroyed upon cycling.

Based on the aforementioned hypothesis, we used PMMA to enhance mechanical

stability of SEI and conducted the AFM to quantify the mechanical stability for YP-Cu. The YP layer exhibits ~ 0.94 GPa after 1 cycle and remain almost constant (~ 1.02 GPa) even after 20 cycles, meaning excellent tolerance to volumetric deformation and enhanced mechanical stability during cycling. These results validate our hypothesis. Therefore, YP-Cu can induce dense lithium deposition and alleviate volume expansion, ultimately enabling a mechanically stable SEI.

In addition, according to the reviewer's suggestion, we also supplemented two other polymers, polyacrylic acid (PAA) and polyvinylidene fluoride (PVDF), to explore whether other polymers could exhibit similar effects to PMMA. As shown in Fig. R21, the YP2 layer with PAA polymer displays ~ 0.89 GPa after 1 cycle and remain almost constant (~ 0.95 GPa) even after 20 cycles, indicating excellent tolerance to volumetric deformation and enhanced mechanical stability upon cycling. Moreover, the carbonyl groups in PAA also exhibit strong interactions with Li^+ ,⁴⁹ guiding uniform deposition and contributing to a dense deposition morphology. For the YP3 layer containing PVDF polymer, the modulus is ~ 1.32 GPa after 1 cycle but sharply decreases to 0.74 GPa after 20 cycles, indicating substantial degradation during cycling and poor cycle stability. Despite PVDF can react with Li to generate the LiF-SEI, which can homogenize the Li^+ flux at the interface. The LiF-rich SEI is brittle and exhibits poor mechanical stability, leading to rupture of SEI and huge volume fluctuation during repeated cycles. In this study, PAA demonstrates effects similar to PMMA, while PVDF does not. Therefore, the polymer chain segments play a crucial role in promoting the uniform deposition of Li^+ and enhancing the mechanical stability of the SEI.

Fig. R21. The modulus distribution for the (a) YP2 layer with PAA and (b) YP3 layer with PVDF of YP-Cu electrode after 1 cycle and 20 cycles.

References:

17. Zhou, J., et al. A new high ionic conductive gel polymer electrolyte enables highly stable quasi-solid-state lithium sulfur battery. *Energy Storage Mater.* **22**, 256-264 (2019).
18. Zhou, Z., et al. A robust, highly stretchable ion-conductive skin for stable lithium metal batteries. *Chem. Eng. J.* **396**, 125254 (2020).
47. Wang, W.-W., et al. Evaluating Solid-Electrolyte Interphases for Lithium and Lithium-free Anodes from Nanoindentation Features. *Chem* **6**, 2728-2745 (2020).
48. Lopez, J., et al. Designing polymers for advanced battery chemistries. *Nat. Rev. Mater.* **22**, 256-264 (2019).
49. Li, N.-W., et al. A flexible solid electrolyte interphase layer for long-life lithium metal anodes. *Angew. Chem. Int. Ed.* **57**, 1505-1509 (2018).

REVIEWER COMMENTS

Reviewer #1 (Remarks to the Author):

Authors' efforts in carrying out additional experiments to address the reviewers' comments are appreciated. But there are still concerns:

1. Author did not address my previous comment on Y-doping satisfactorily. Authors' response suggests they may have some misunderstanding of the comments. In their response, they focus on defending "Y-doping did not change the structure of Li". Obviously this is not what I meant in the comment. What I am asking is to show evidence to support the claim that Y is doped into the lithium crystal structure. This is important because Y being doped into lithium is the basis of other important arguments like the surface energy calculation.

2. On the Li||Cu CE. If authors want to show the benefit of using YP, they should compare the CE of Li||YP-Cu with that of the baseline Li||Cu. The same electrolyte 4M LiFSI in DME should be used for both cases. Figure R2 is not providing the state-of-the-art CE in Li||Cu cells. These papers (Nature Energy 2022, 7, 94–106; Nature Energy, 2020, 5, 526–533; Chem, 2023, 9, 650-664) are some of the examples that have CE above 99.5%. The CE number for ref1 in Figure R1 is not accurate.

Editorial note: Reviewer #1 was additionally asked to comment in the place of Reviewer #3. The additional comments are as follows:

Reviewer #3's concern includes novelty and mechanism. Novelty probably can be argued. The concern over mechanism clarification is valid and I share the same concern. R3 asked to provide other yttrium compound data to support the claim about the mechanism. Authors showed YCl₃ data but this is dubious because Cl⁻ is a well known headache for the cathode current collector—Al foil. It corrodes Al very aggressively. If authors can show evidence they indeed use YCl₃ (maybe like XPS and elemental mapping) and it also worked well, I think it counts as convincing evidence to support that Y plays the key role in changing Li plating behavior. This finding would be important and the paper can be accepted.

Reviewer #2 (Remarks to the Author):

The comments have been addressed by the authors and publication is recommended.

Response to Reviewer #1

Reviewer #1: Authors' efforts in carrying out additional experiments to address the reviewers' comments are appreciated. But there are still concerns:

Response:

Thanks a lot for your comments. We would like to express our sincere appreciation for your careful reading and helpful comments. In response to your insightful suggestions, we have provided new experimental evidence, including X-ray diffraction (XRD) patterns and Cryo-transmission electron microscopy (Cryo-TEM), in the revised manuscript to demonstrate the Y doping into the Li metal, as detailed in *Reviewer Comments 1*. Additionally, we supplemented the Aurbach Coulombic efficiency (CE) test of Li||YP-Cu and Li||Bare Cu using the same electrolyte 4M LiFSI in DME, as detailed in *Reviewer Comments 2*. We hope that these revisions to the manuscript address your concerns.

Reviewer Comment 1: Author did not address my previous comment on Y-doping satisfactorily. Authors' response suggests they may have some misunderstanding of the comments. In their response, they focus on defending "Y-doping did not change the structure of Li". Obviously, this is not what I meant in the comment. What I am asking is to show evidence to support the claim that Y is doped into the lithium crystal structure. This is important because Y being doped into lithium is the basis of other important arguments like the surface energy calculation.

Response:

We are thankful to the reviewer's comment. Based on your suggestion, we supplemented the XRD patterns and Cryo-TEM images to further demonstrate the Y doping into the Li metal in the revised manuscript (page 9, line 11-17).

Firstly, we conducted XRD patterns of YP-Li electrodes with varying YP concentrations from 0.1 wt.% to 2.0 wt.% (Fig. R1a). In comparison to bare Li, the (110) peak of YP-Li exhibits a noticeable shift towards lower angle (Fig. R1b), which is attributed to the doping effect of Y with a larger radius.^{1, 2} Furthermore, as the YP content increases, the degree of shift in the (110) peak also increases, suggesting a

higher level of Y doping into the Li metal (Fig. R1). Additionally, the enlarged views of XRD patterns in the range of 36.0 to 37.4° after different cycles for bare Li and YP-Li electrodes (Supplementary Fig. 10) exhibit a low-angle shift in the (110) peak of YP-Li compared with bare Li electrodes, as shown in Fig. R2. This is also a result of Y doping, which has a larger radius than Li.

Furthermore, the morphology and structure of YP-Li was further investigated via Cryo-TEM. HRTEM images reveal a lattice space of 0.24 nm corresponding to the (110) plane of Li crystal (Supplementary Fig. 6a-b), as verified by the corresponding local Fourier transform images (inset of Supplementary Fig. 6c). Energy-dispersive X-ray spectroscopy (EDS) elemental mapping results (Supplementary Fig. 6d) indicate the existence of Y element in the lithium metal, providing additional confirmation of Y doping into the Li metal.^{1,2}

At last, X-ray photoelectron spectroscopy (XPS) of YP-Cu electrodes after Li plating was further investigated to confirm the Y doping into Li metal. The binding energy at ~157.8 eV in the Y 3d_{5/2} spectra (Supplementary Fig. 5e) and 55.8 eV in the Li 1s spectra (Supplementary Fig. 5f) correspond to Y⁰ and LiF,³ respectively, indicating the reduction of YF₃ to form Y⁰ and LiF during Li plating. Therefore, the combined results from XRD, Cryo-TEM, and XPS suggest that Y has been successfully doped into Li metal.

Fig. R1. (a) XRD patterns for bare Li, 0.1 wt.% YP-Li, 0.5 wt.% YP-Li, 1 wt.% YP-Li, and 2 wt.% YP-Li after 10 cycles, and (b) the enlarged view in the 2θ range of 36.0-37.4°.

Supplementary Fig. 10. XRD patterns of the (a) bare Li and (b) YP-Li after different cycles.

Fig. R2. The enlarged view in the 2θ range of 36.0-37.4° of bare Li and YP-Li after different cycles.

Supplementary Fig. 6. (a) Low-resolution Cryo-TEM images of YP-Cu electrodes. (b) Enlarged TEM image and (c) corresponding HR-TEM and FFT images (inset). (d) EDS elemental mappings of Y element.

Supplementary Fig. 5e-f. XPS spectra of (e) Y 3d_{5/2} and (f) Li 1s for the YP-Cu electrode after Li plating.

References:

1. Huang, S., et al. Y-doped Li₈ZrO₆: A Li-ion battery cathode material with high capacity. *J. Am. Chem. Soc.* **137**, 10992-11003 (2015).
2. Ou, X., et al. Enabling high energy lithium metal batteries via single-crystal Ni-rich cathode material co-doping strategy. *Nat. Commun.* **13**, 2319 (2022).
3. Xie, Y., et al. Surface modification using heptafluorobutyric acid to produce highly stable Li metal anodes. *Nat. Commun.* **14**, 2883 (2023).

Reviewer Comment 2: On the Li||Cu CE. If authors want to show the benefit of using YP, they should compare the CE of Li||YP-Cu with that of the baseline Li||Cu. The same electrolyte 4M LiFSI in DME should be used for both cases. Figure R2 is not providing the state-of-the-art CE in Li||Cu cells. These papers (*Nature Energy* 2022, 7, 94–106; *Nature Energy*, 2020, 5, 526–533; *Chem*, 2023, 9, 650-664) are some of the examples that have CE above 99.5%. The CE number for ref1 in Figure R2 is not accurate.

Response:

Thanks a lot for the reviewer's comments. Following your suggestion, we supplemented the Aurbach CE test⁴ on Li||YP-Cu and Li||Bare Cu cells using the same electrolyte of 4 M LiFSI in DME. This approach provides a more comprehensive evaluation of the efficiency of Li cycling on a Li metal, further confirming the advantages of the YP-Cu anode. Specifically, YP-Cu demonstrates a significant enhancement in CE (99.52%) compared with bare Cu (99.01%) (Fig. R3). In addition, we have added the state-of-the-art CE (*Nature Energy* 2022, 7, 94–106; *Nature Energy*, 2020, 5, 526–533; *Chem*, 2023, 9, 650-664) of the Li||Cu cells, as shown in Fig. R4. These papers have been of great assistance and inspiration to our research, which have

been cited in the *Introduction section* in the revised manuscript (Ref 10-12, page 3, line 19).

The performance of asymmetric Li||Cu cells is affected by various factors, including areal capacity, current density, and electrolyte dosage, etc. Achieving long cycle life of Li||Cu cells **under high areal capacity and high current density conditions** remains a significant challenge.^{5, 6} In our study, the asymmetric Li||Cu cells were tested under approaching practical conditions, including high deposition capacity of 4 and 8 mAh cm⁻², high current density of 1 mA cm⁻², and lean electrolyte dosage of 70 μ L. In comparison, the Li||Cu cells of these papers (*Nature Energy* 2022, 7, 94-106; *Nature Energy*, 2020, 5, 526-533; *Chem*, 2023, 9, 650-664) were conducted under the low areal capacity of 1 mAh cm⁻² and low current density of 0.5 mA cm⁻². Under high areal capacity conditions (≥ 4 mAh cm⁻²), the volume expansion of the Li metal anode is exacerbated,⁷ which leads to repeated fracturing and regeneration of the SEI of Li metal batteries during extended cycling. Such phenomenon induces uneven Li deposition, exacerbating dendrite growth, and ultimately resulting in poor cycling stability of Li||Cu cells. Therefore, YP-Cu exhibits more impressive cycling performance than those of previous reports, especially for the high deposition capacities (≥ 4 mAh cm⁻²). The enhanced and more stable CEs mean a highly reversible reaction, contributing to superior interface stability of the YP-Cu anode.

For *Ref1* (*Nature Energy* 2023, 8, 725-735) in Fig. R2 in the last version, the cell with bilayer/P-F SEI maintains a high CE (>99.5%) within 400 cycles at a current density of 1.0 mA cm⁻² and a Li plating/stripping capacity of 1.0 mAh cm⁻². Even when the capacity increases to 3.0 mAh cm⁻², the cell with bilayer/P-F SEI maintains a high CE (>99.4 %) within 100 cycles.⁸

Fig. R3. Aurbach measurement of Li metal CE in the Li||YP-Cu and Li||Bare Cu half cells using the same electrolyte of 4M LiFSI in DME.

Fig. R4. The comparison of the Coulombic efficiencies and area capacities of the Li||Cu cells in this work and literatures.

References:

4. Adams, B. D., et al. Accurate determination of coulombic efficiency for lithium metal anodes and lithium metal batteries. *Adv. Energy Mater.* **8**, 1702097 (2018).
5. Niu, C., et al. Self-smoothing anode for achieving high-energy lithium metal batteries under realistic conditions. *Nat. nanotechnol.* **14**, 594-601 (2019).
6. Niu, C., et al. High-energy lithium metal pouch cells with limited anode swelling and long stable cycles. *Nat. Energy* **4**, 551-559 (2019).
7. Fang, C., et al. Pressure-tailored lithium deposition and dissolution in lithium metal batteries. *Nat. Energy* **6**, 987-994 (2021).
8. Zhang, Q.-K., et al. Homogeneous and mechanically stable solid–electrolyte interphase enabled by trioxane-modulated electrolytes for lithium metal batteries. *Nat. Energy* **8**, 725-735 (2023).

Response to Reviewer #3

Reviewer #3: Reviewer #3's concern includes novelty and mechanism. Novelty probably can be argued. The concern over mechanism clarification is valid and I share the same concern. R3 asked to provide other yttrium compound data to support the claim about the mechanism. Authors showed YCl_3 data but this is dubious because Cl is a well-known headache for the cathode current collector-Al foil. It corrodes Al very aggressively. If authors can show evidence, they indeed use YCl_3 (maybe like XPS and elemental mapping) and it also worked well, I think it counts as convincing evidence to support that Y plays the key role in changing Li plating behavior. This finding would be important and the paper can be accepted.

Response:

We are grateful with the reviewer's comment. In our work, the YCl_3 coating amount is small, only 0.1 wt.%. In response to your suggestion, we added leakage current density and SEM images of YCl_3 -Al and Al foil after holding the cell at 4.7 V for 12 h to assess the extent of corrosion. As shown in Fig. R5, both YCl_3 -Al and Al exhibit similar and low leakage current density, consistent with the SEM images revealing negligible corrosion on the Al surface. This phenomenon further illustrates that due to the low coating amount of YCl_3 , for just 0.1 wt.%, the corrosion can be negligible.

Furthermore, we supplemented SEM-EDS images and XPS spectra of YP1-Cu (YCl_3) electrode before and after cycling to demonstrate the effect of YCl_3 . The uniform distribution of C, O, Y, and Cl elements proves that YP1 is evenly covered on the Cu foil (Fig. R6). XPS was further carried out for YP1-Cu electrode before and after cycling to investigate the stability of the YP1 layer, as shown in Fig. R7. Before cycling, the binding energy at 284.6, 286, and 288.8 eV refer to the C-C, C-O, and C=O bonds of PMMA, respectively. The binding energy at ~ 162.3 eV in the Y $3d_{5/2}$ spectra and ~ 199.8 eV in the Cl 2p is assigned to YCl_3 , signifying the successful coating of YP1 on the Cu foil. After Li deposition, the binding energy at ~ 160.5 eV in the Y $3d_{5/2}$ spectra and ~ 196.3 eV in the Cl 2p spectra correspond to Y and LiCl,³ indicating the reduction of YCl_3 to form Y and LiCl during Li plating.

Fig. R5. (a) Leakage current density and (b-c) SEM images of YCl₃-Al and Al electrodes after being held at 4.5 V (vs. Li+/Li) for 12 h.

Fig. R6. The EDS element mappings of C, O, Y, and Cl in the YP1-Cu electrode.

Figure R7. XPS spectra of C 1s, Y 3d_{5/2}, and Cl 2p for the YP1-Cu before and after cycling.

Reference:

- Xie, Y., et al. Surface modification using heptafluorobutyric acid to produce highly stable Li metal anodes. *Nat. Commun.* **14**, 2883 (2023).

REVIEWERS' COMMENTS

Reviewer #1 (Remarks to the Author):

All reviewers' comments addressed and the paper can be published as it is.